

# A review of the common crab genus *Macromedaeus* Ward, 1942 (Brachyura, Xanthidae) from China Seas with description of a new species using integrative taxonomy methods

Ziming Yuan[1,2], Wei Jiang[1,3] and Zhongli Sha[1,3,4]

[1] Laboratory of Marine Organism Taxonomy and Phylogeny, Institute of Oceanology, Chinese Academy of Sciences, Qingdao, China
[2] University of Chinese Academy of Sciences, Beijing, China
[3] Shandong Province Key Laboratory of Experimental Marine Biology, Institute of Oceanology, Chinese Academy of Sciences, Qingdao, China
[4] Laboratory for Marine Biology and Biotechnology, Qingdao National Laboratory for Marine Science and Technology, Qingdao, China

## ABSTRACT

*Macromedaeus* is one of the most common xanthid genera in shallow waters of the Indo-West Pacific. In this study, we describe a new species, *Macromedaeus hainanensis* sp. nov., and report on two newly recorded species, *M. quinquedentatus* (*Krauss, 1843*) and *M. orientalis* (*Takeda & Miyake, 1969*) from Hainan Island, South China Sea. *M. hainanensis* is most related to *M. distinguendus* (*De Haan, 1833-1850*) and *M. orientalis* on the carapace shape and granular appearance, but can be distinguished by unique morphological characteristics especially its front, pereopods and male first gonopod. Taxonomic identities of the six *Macromedaeus* species recorded from China seas are discussed, and a phylogenetic analyzation is performed on *Macromedaeus* and related taxa based on three mitochondrial and two nuclear markers (12S, 16S, COI, H3, 18S). Integrated taxonomic evidence is used to support the taxonomic status of each species.

## INTRODUCTION

The xanthid crab genus *Macromedaeus* (*Ward, 1942*) currently includes six described species found from the Indo-West Pacific are often common and abundant members in intertidal fauna of rocky seashores (*Ng, Guinot & Davie, 2008*). The genus was established with *M. punctatus* (*Ward, 1942*) as the type species, and he also attached *Xantho nudipes Milne-Edwards, 1867* to *Macromedaeus*. However, *M. punctatus* was considered synonymous with *M. nudipes* for the slight differences, and now *M. nudipes* is taken as the type species of *Macromedaeus* (*Ward, 1942*; *Guinot, 1968*; *Guinot, 1971*; *Serène, 1984*).

*Guinot (1968)* transferred five species from *Xantho* Leach, 1814 or *Leptodius* A. Milne-Edwards, 1863 to *Macromedaeus*. *Macromedaeus* differs from its most related genus *Leptodius* by: antennary basal segment stout and straight in *Macromedaeus*, while it slender

Corresponding author
Zhongli Sha, shazl@qdio.ac.cn

and inclined in *Leptodius*; antennular fossa straighter and shorter in *Leptodius*; third maxillipeds stouter, merus with a protruding external angle in *Leptodius* and the first gonopod of male (G1) of *Macromedaeus* having a stout apical lobe, while in *Leptodius* the G1 having a slender apical lobe with prominent mushroom-like or tongue-like extensions (*Guinot, 1968*).

*Macromedaeus* currently consists of six species: *M. crassimanus* (A. (*Milne-Edwards, 1867*), *M. demani* (*Odhner, 1925*), *M. distinguendus* (*De Haan, 1833-1850*), *M. nudipes* (A. (*Milne-Edwards, 1867*), *M. quinquedentatus* (*Krauss, 1843*) and *M. voeltzkowi* (*Ng, Guinot & Davie, 2008*; *Lenz, 1905*), and three of them (*M. distinguendus*, *M. crassimanus* and *M. demani*) have been reported from China seas (*Liu, 2008*). *Microcassiope orientalis* (*Takeda & Miyake, 1969*) should be transformed into *Macromedaeus* (*Yamaguchi, Takeda & Tokudome, 1976*; *Takeda, 1977*; *Maenosono, 2021*), but this suggestion was not accepted widely (*Ng, Guinot & Davie, 2008*; *Lee, 2012*).

During a recent collection from Hainan Island, several individuals referable to *Macromedaeus* were collected. Some of them exhibit distinctive characters differentiating them from other described *Macromedaeus* species, so they are identified as a new species herein. *M. quinquedentatus* is newly recorded from China seas. Additionally, *Microcassiope orientalis* is also founded from China seas firstly. A review of *Macromedaeus* is performed using integrative taxonomy methods, and an identification key for *Macromedaeus* species is provided.

## MATERIALS AND METHODS

The specimens were collected from China Seas by intertidal collections from 1950 to 2020. They were stored in 70% ethanol and deposited in the Marine Biological Museum, Chinese Academy of Sciences (MBMCAS), Qingdao, China.

Morphological characteristics were observed using ZEISS Stemi 2000-c and ZEISS Stemi SV 11 Apo stereo microscope, and Nikon Eclipse Ci-L microscope. Photographs were taken using Canon EOS 6D camera with Canon EF 100 mm and Canon MP-E 65 mm lens or using Nikon D800 camera with Nikon AF-S 105 mm lens.

The morphological terms followed that used by *Dana (1852)* and *Serène (1984)*, and the following abbreviations were used in the text: CW (maximum carapace width); CL (median carapace length); ab1-6 (abdominal somites 1–6); st1-8 (thoracic sternites 1–8); P1-5 (pereopods 1–5); G1 (first gonopod of male); G2 (second gonopod of male); 1-2F (frontal regions 1–2); 1-4M (medial regions 1–4); 1-6L (antero-lateral regions 1-6); 1-3R (postero-lateral regions 1–3); 1-2P (posterior regions 1–2).

Molecular phylogenetic analyses were performed on *Macromedaeus* species and related taxa to understand their phylogenetic position. Genomic DNA was extracted from muscle tissue using the OMEGA EZNA Tissue DNA Kit (USA) following the manufacturer's protocol. Molecular characters were obtained from three mitochondrial and two nuclear markers, which were mitochondrial 12S rRNA (12S, approximately 363 bp), 16S rRNA (16S, approximately 521 bp), cytochrome oxidase I (COI, approximately 658 bp), nuclear histone H3 (H3, approximately 328 bp) and 18S rRNA (18S, approximately 644 bp). Markers were

amplified via polymerase chain reaction (PCR) with primers 12sf and 12s1r for 12S (*Buhay et al., 2007*), 16Sar and 16Sbr for 16S (*Palumbi, 1996*), dgLCO-1490 and dgHCO-2198 for COI (*Meyer, 2015*), 18S-B and 18S-O for 18S (*Medlin et al., 1988*; *Apakupakul, Siddall & Burreson, 1999*), and Hex-AF and Hex-AR for H3 (*Svenson & Whiting, 2004*).

PCR was performed in 25 μl volumes containing: 1 μl (3–200 ng) of genomic DNA as template, 1 μl (10 pM) of each primer, 12 μl of 2×PCR Mix (Dongsheng Biotech, Guangzhou, China) and 10 μl ultrapure water. Reactions were carried out as following steps: initial denaturation at 95 °C for 5 min; 35 cycles for denaturation at 95 °C for 30 s, annealing at 60 °C (12S), 48 °C (16S), 58 °C (18S), 48 °C (COI), 66 °C (H3) for 45 s, extension at 72 °C for 45 s, and final extension at 72 °C for 10 min.

The obtained sequences were edited using Lasergene and aligned by MEGA version 6 (*Tamura et al., 2013*). Alignments were concatenated by SequenceMatrix 1.8 (*Vaidya, Lohman & Meier, 2011*). In the final dataset, including gaps, there were total 363 bp for the 12S, 534 bp for the 16S, 499 bp for the 18S, 524 bp for the COI and 306 for H3 dataset.

Maximum Likelihood (ML) and Bayesian Inference (BI) analyses were implemented in phylogenetic studies. The best-fit model of evolution for each dataset was identified using jModeltest 0.1.1 under the Akaike information criterion (AIC) (*Posada, 2008*). BI analyses were carried out using MrBayes 3.2.7 (*Huelsenbeck & Ronquist, 2001*). A Markov ChainMonte Carlo (MCMC) algorithm with two runs of four chains each was run for 1,000,000 generations with trees sampled every 500 generations (2,000 trees sampled). The first 500 trees were discarded and the posterior probabilities were estimated for the remaining samples. The ML analyses were performed online at W-IQ-TREE (*Jana et al., 2016*). Clade support was assessed with 1,000 ML bootstrap replications. The genetic divergences of COI between and within the Macromedaeus species were constructed in MEGA version 6 (*Tamura et al., 2013*).

The electronic version of this article in Portable Document Format (PDF) will represent a published work according to the International Commission on Zoological Nomenclature (ICZN), and hence the new names contained in the electronic version are effectively published under that Code from the electronic edition alone. This published work and the nomenclatural acts it contains have been registered in ZooBank, the online registration system for the ICZN. The ZooBank LSIDs (Life Science Identifiers) can be resolved and the associated information viewed through any standard web browser by appending the LSID to the prefix http://zoobank.org/. The LSID for this publication is: urn:lsid:zoobank.org:pub:07834F78-CD9D-4516-A92C-7FC5B680A7F9. The online version of this work is archived and available from the following digital repositories: PeerJ, PubMed Central SCIE and CLOCKSS.

# RESULTS

## Taxonomy

**Family Xanthidae MacLeay, 1838**
**Subfamily Xanthinae MacLeay, 1838**
**Genus *Mac romedaeus* Ward, 1942**
***Macromedaeus hainanensis* sp. nov.**

## Material examined

**Type material. Holotype.** One male (6.54 × 4.34 mm); Nov. 12, 2016; Mulan Bay, Wenchang, Hainan, Junlong Zhang, Yang Li, Shuqian Zhang et al. coll. (MBM286988). **Paratypes.** One female (14.69 × 9.21 mm); Nov. 3, 1990; Qukou, Haikou, Hainan (MBM160965); one female; Nov. 12, 2016; Mulan Bay, Wenchang, Hainan; Junlong Zhang, Yang Li, Shuqian Zhang et al. coll. (MBM286989) one female (6.94 × 4.54 mm); Nov. 12, 2016; Mulan Bay, Wenchang, Hainan; Junlong Zhang et al. coll. (MBM286990); one male (7.14 × 4.86 mm), two females (5.02 × 3.39 mm; 4.44 × 3.35 mm); Nov. 12, 2016; Mulan Bay, Wenchang, Hainan , Junlong Zhang et al. coll. (MBM286991); one male (7.56 × 4.75 mm); Nov. 29, 2007; Linchang Reef, Hainan, Xianqiu Ren coll. (MBM282500); two females (9.44 × 5.99 mm, 5.53 × 3.78 mm); Apr. 13, 2008; Linchang Reef, Danzhou, Hainan; Wei Jiang, Yang Li coll. (MBM286992).

## Description of the holotype

Carapace (Figs. 1A, 1B, 2A) transversely ovate, width about 1.5 times length, dorsal slightly convex; dorsal surface covered with granules, more or less arranged in lines; regions well defined in anterior 2/3, especially 2M, 3M, 5L and 6L, subhepatic and pterygostomian regions (Fig. 1C) granular. Front (Figs. 1B, 2C) double-rimmed, both granular, not produced, slightly deflexed, about 0.3 times carapace width, divided into two lobes by a v-shape mesial notch, margin of frontal lobes nearly straight, external part advanced. Anterolateral border convex, divided into four teeth except exorbital angle, separated from each other by wide v-shape notches, first low, triangular, not advanced, second widest, anterior margin shorter than posterior, third most advanced, anterior margin shorter than posterior, fourth small but distinct. Posterolateral margin slightly longer than anterolateral margin. Posterior margin slightly convex.

   Orbits suboval (Figs. 1B, 2A), orbital region granular; supraorbital margin with two fissures, dorsal inner orbital angle separated from front by a deep notch; infraorbital tooth blunt. Eyestalk stout, covered with granules. Antennular fossae subrectanglar; antennules folding transversely. Basal segment of antenna subrectanglar, flagellum entering orbital hiatus, tip reaching exorbital angle. Epistome broad, posterior margin with a low and blunt mesial prominence. Endostome with oblique ridges posteriorly.

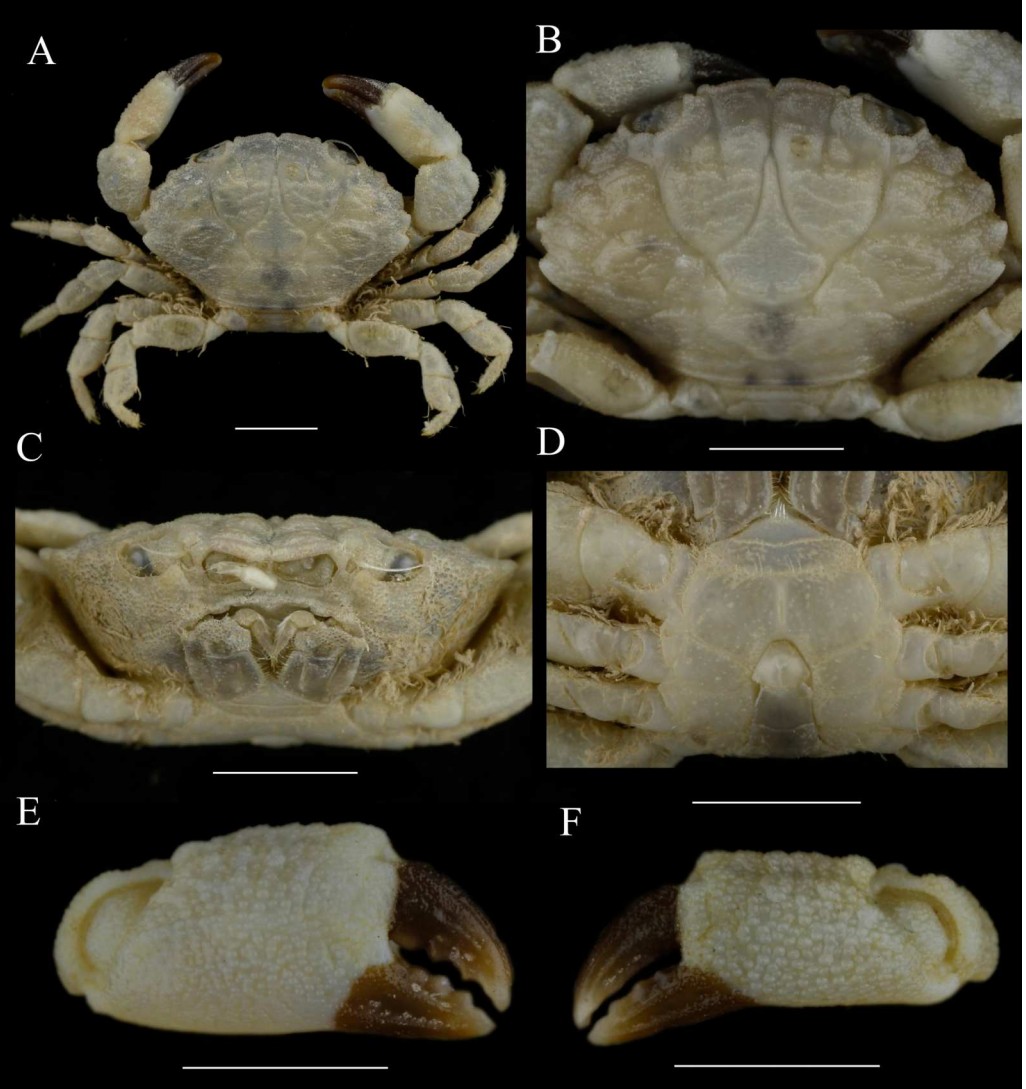

**Figure 1** **Holotype of *Macromedaeus hainanensis*. sp. nov., male, 6.54 × 4.34 mm (MBM286988).** (A) Overall, dorsal view. (B) Carapace, dorsal view. (C) Third maxillipeds and pterygostomian region, anterior view. (D) Thoracic sternites and abdomen. (E) Right cheliped, outer view. (F) Left cheliped, outer view. Scale bars = 2 mm.

Third maxillipeds (Figs. 1C, 2B) completely covering buccal orifice; merus subquadrate, granular, anterolateral angle slightly produced; ischium subrectangular, about two times as long as merus, surface nearly smooth, sulcate submedially; exopod broad, subdistal portion with a rounded projection beneath inner margin.

Male thoracic sternum (Fig. 1D) slightly granular, st1, two fused, suture between st2, 3 distinct, backward convex; st3 and st4 almost fused, suture only present on lateral edges, and continue by a feeble but visible transverse depression in middle part; st4 broad, with a distinct median longitudinal groove; press-button on posterior of st5.

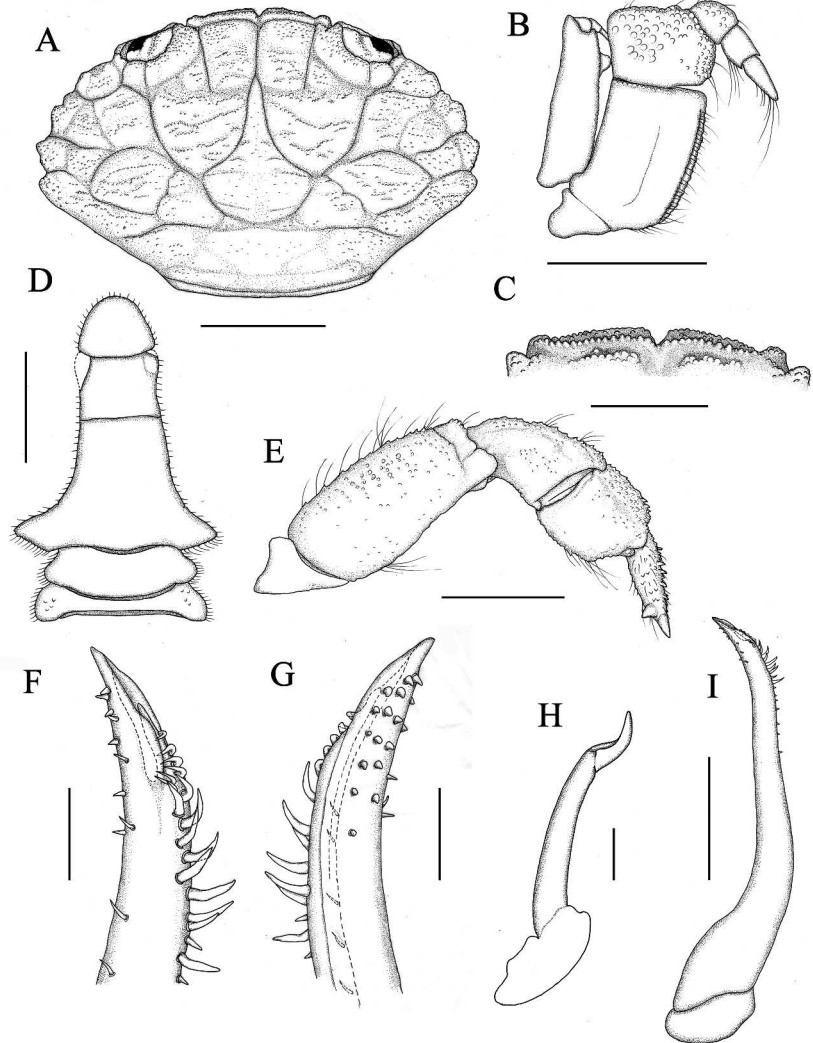

**Figure 2** Holotype of *Macromedaeus hainanensis* sp. nov., male, 6.54 × 4.34 mm (MBM286988). (A) Carapace, dorsal view. (B) Third maxillipeds, external view. (C) Front, dorsal view. (D) Abdomen. (E) P5, dorsal view. (I) G1, left, internal view. (F) Same, distal portion, internal view. (G) Same, external view. (H) G2, right, external view. Scale bars: (A) 2 mm. (B–E) = 1 mm. (F–H) = 0.1 mm. (I) = 0.5 mm.

Chelipeds (Figs. 1E, 1F) unequal, robust; merus short, with setae on inner edge; carpus robust, with granules and finely areoles on dorsal surface, internal angle blunt; palm uniform granular, with a groove on outer surface near dorsal; fingers short and robust, brown pigmented, nearly not extent to palm, cutting margins with 4 blunt teeth on both fingers, an enlarged blunt tooth on dactylus subbasal part of large cheliped; dactylus shorter than superior margin of palm, with two dorsal longitudinal grooves; fixed finger with two granular lines on outer surface; finger tips spoon-shaped, with a tuft of setae.

Ambulatory legs (Figs. 1A, 2E) short, slightly depressed, edges with long setae, P3 and P4 longest, P5 (Fig. 2E) shortest; merus with granules on dorsal surface; carpus curved, covered with granules; propodus subrhombic, with granules; dactylus

with spiny granules and setae, a sharp and strong subterminal tooth on dactylus posterior margin of the last three ambulatory legs, larger on posterior dactylus, tip claw-shaped; dactylo-propodal articulation with distinct lock composed of rounded prolongation on propodus distal lateral margin sliding against prominent button on dactylus proximal lateral margin; P5 dactylus tip nearly straight, slightly forward pointed.

Male abdomen (Figs. 1D, 2D) with ab1 and ab2 broad, subtrapezoidal; ab3-5 completed fused; ab6 subquadrate, width slightly larger than length, enlarger on subdistal; telson subtriangular, angles rounded, basal width slightly larger than median length.

G1 (Figs. 2F, 2G, 2I) long and slender, slightly curving laterally; distal tip with a sharp and elongate apical lobe wrapping the subdistal lobe; subdistal part with about 17 subterminal spines, distal 6 of them on the prominent subdistal lobe; outer ventral surface of subdistal part with little spines. G2 slightly sigmoidal in shape, about 1/4 length of G1.

**Female morphology.** The female (Fig. 3) is similar to male in morphology generally, but there are still some differences. The largest female specimen shows a more prominent and narrower front (front about 0.27 times CW; 0.3 times in the male holotype) and a wider carapace (CW/CL = 1.6; 1.5 in the male holotype) (Figs. 3A, 3B) compared with the male holotype which may be caused by the growth of the individuals. In females, the abdomen (Fig. 3D) is wider and oval with seta on the margin, and pleonal somites not fused. The chelipeds (Figs. 3E, 3F) are equal, and palms are slender.

**Etymology.** The species is named after its type locality, Hainan Island, Hainan province, China.

**Distribution.** Presently only known from the type locality, Hainan Island.

**Remarks.** *Macromedaeus hainanensis* sp. nov. is most similar to *M. distinguendus* for having four triangular anterolateral teeth and granular carapace and to *M. orientalis* for having smaller body size and the same front.

*M. hainanensis* sp. nov. differs from *M. distinguendus* by: carapace more granular, regions relatively flat, with granular lines but not form produced crests (carapace regions more prominent, with obvious granular crests in *M. distinguendus*); anterolateral teeth narrower, apices more projecting, forming an acute angle (anterolateral teeth broader, apices slightly projecting, forming an obtuse angle in *M. distinguendus*); the front double-rimmed, at least 0.27 times CW, not produced, margin of frontal lobes nearly straight (not double-rimmed, about 0.22 times CW, produced, margin leaning in *M. distinguendus*); ambulatory legs carpus without distinct crest, dactylus with a strong subterminal tooth in the last three legs. (carpus with three crests, dactylus without distinct subterminal tooth in *M. distinguendus*); P5 dactylus tip nearly straight, slightly pointed forward (P5 dactylus tip curved backward in *M. distinguendus*); G1 stout with a sharp and elongate apical lobe, subdistal lobe short, with 6 curving spines on the prominent lube. (G1 slender, with blunt apical lobe, subdistal lobe longer, with about 11 curving spines on the prominent lube *in M. distinguendus*) and *M. hainanensis* sp. nov. obviously smaller than the latter (the largest specimens of *M. hainanensis* sp. nov. having a 14.69 mm CW, and the largest *M. distinguendus* having 30.58 mm CW).

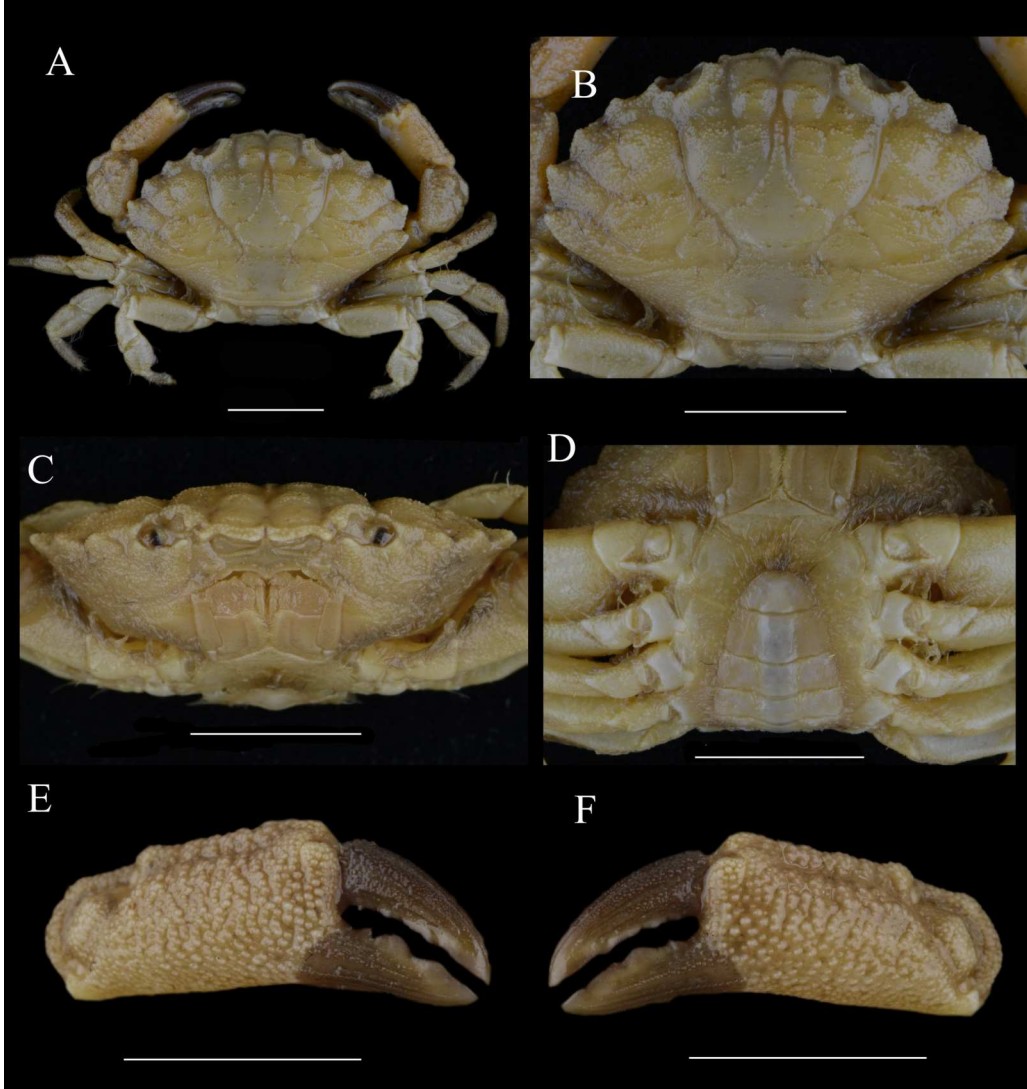

**Figure 3** **Paratype of *Macromedaeus hainanensis* sp. nov., female, 14.69 × 9.21 mm (MBM160965).** (A) Overall, dorsal view. (B) Carapace, dorsal view. (C) Third maxillipeds and pterygostomian region, anterior view. (D) Thoracic sternites and abdomen. (E) Right cheliped, outer view. (F) Left cheliped, outer view. Scale bars = 5 mm.

*M. hainanensis* sp. nov. differs from *M. orientalis* by: P5 stout, dactylus tip nearly straight, slightly pointed forward (P5 slender, propodus and dactylus prolonged, with a longer dactylus tip curved backward in *M. orientalis*); chelipeds palm outer surface with denser granules, more uniform in size (it with sparser but more prominent granules, the larger one with two longitudinal grainy lines in *M. orientalis*); unmoved finger of male chelipeds with brown color nearly not spread to palm. (it spread to palm inner and outer surface in *M. orientalis*); G1 slender, apical lobe not upturned, the ventral rim more prominent than the dorsal rim (G1 stout, apical lobe slightly upturned, the dorsal rim more prominent than the ventral rim in *M. orientalis*).

### *Macromedaeus orientalis* (*Takeda & Miyake, 1969*)

*Microcassiope orientalis Takeda & Miyake, 1969*: 202–205, figs. 2, 3; *Ng, Guinot & Davie, 2008*: 203 (list); *Lee, 2012*: 160–164, figs. 122–124.

*Macromedaeus orientalis Yamaguchi, Takeda & Tokudome, 1976*: 37, fig 2; *Takeda, 1977*: 84; *Lee & Ko, 2008*: 17–19, fig. 1; *Maenosono, 2021*: 10, figs. 1h–i, figs. 7a–f, 8f.

**Material examined:** One female; Nov. 21; 2016, Hainan, subtidal zone 9–10 m; Junlong Zhang coll, (MBM286993); two males; Mar. 19, 1992; Hainan (MBM286994); one female, Nov. 20, 1990; Yalong Bay, Sanya, Hainan (MBM164388); one male; Nov. 20, 1990; Yalong Bay, Sanya, Hainan, (MBM164386).

**Size.** CW: 5.20–8.14 mm, CL: 3.47–5.23 mm.

**Diagnosis.** Carapace (Figs. 4A, 4B) transversely ovate, the breadth is about 1.5 times the length; regions well-defined on anterior 2/3, covered with granules and form rows of transverse line; inner and outer lobules of 2M, the anterior edge of 5L and inner angle of 6L with a tuft of brush-like setae respectively. Front (Fig. 5A) about 1/3 the breadth of carapace, not produced, double-rimmed, both granular, divided into two lobes by a v-shape notch, margin of frontal lobes rounded. supraorbital margin with two fissures. Anterolateral border armed with four teeth except the outer orbital angle, the second tooth largest. Posterolateral border subequal with anterolateral border. Lateral surface of carapace clothed with setae. Antennule situated transversely; orbital hiatus filled by antennal flagellum. Third maxilliped (Fig. 4C) completely covering buccal orifice; merus subquadrate, granulated; ischium subrectangular, with a smooth groove. Thoracic sternites (Fig. 4D) smooth; the suture between st1-2 and 3 distinct, backward convex, the median line of st4 distinct.

Chelipeds (Figs. 4E, 4F) unequal; merus with granules on outer surface and setae on inner edge; carpus armed with a tooth on inner angle, dorsal surface with granules and tubercles; palm covered with sharp granules, more prominent on the smaller cheliped, the large cheliped with two longitudinal grainy lines; fingers black brown, the color of fixed finger extend to palm irregularly in male; dactylus with two longitudinal dorsal grooves, cutting edges of the large cheliped with 4 blunt tooth; finger tips spoon-shaped, with a tuft of setae.

Ambulatory legs (Figs. 4A, 6A) with sharp granular, edges with long setae; dactylus armed with spiny granules and setae, tip claw-shaped; a sharp subterminal tooth on dactylus posterior margin of the last three ambulatory legs, larger on posterior dactylus; dactylo-propodal articulation with distinct lock composed of rounded prolongation on propodus distal lateral margin sliding against enlarged prominent button on dactylus proximal lateral margin; P5 (Fig. 6A) slender, propodus and dactylus prolonged, dactylus tip prolonged, curved backward.

Abdominal somites (Fig. 4D) 3–5 completely fused in male. G1 (Figs. 7A, 7E, 7I) stout, slightly curving laterally; distal tip with a apical lobe slightly upturned on the tip and wrapping the subdistal lobe, the apical lobe dorsal rim more prominent than the ventral rim; subdistal part with about 16 curving subterminal spines, distal 5 on the prominent subdistal lobe; outer ventral surface of subdistal part with little spines.

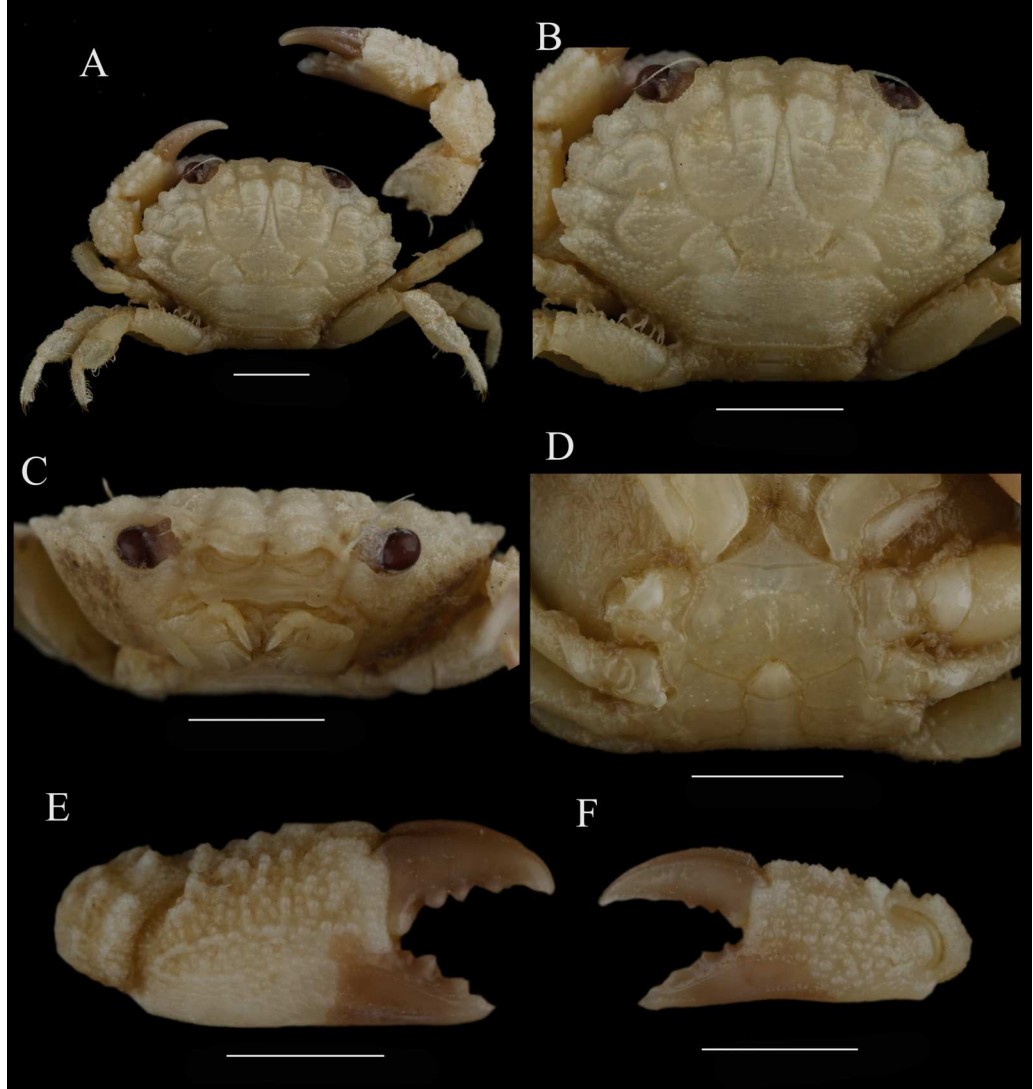

**Figure 4** *Macromedaeus orientalis* (*Takeda & Miyake, 1969*), male, 6.85 × 4.43 mm (MBM286994). (A) Overall, dorsal view. (B) Carapace, dorsal view. (C) Third maxillipeds and pterygostomian region, anterior view. (D) Thoracic sternites and abdomen. (E) Right cheliped, outer view. (F) Left cheliped, outer view. Scale bars = 2 mm.

**Type locality.** Munakata-Oshima Islet, Fukuoka Prefecture, Japan.

**Distribution.** Hainan Island; southeast of Korea, Japan.

**Remarks:** *M. orientalis* was first referred to *Microcassiope Guinot, 1967*. *Yamaguchi, Takeda & Tokudome (1976)* suggested that the species should be transferred to *Macromedaeus Ward, 1942*, but *Ng, Guinot & Davie (2008)* and *Lee (2012)* still listed the species under *Microcassiope*. *M. orientalis* is listed under *Macromedaeus* in present study for *M. orientalis* sharing similar G1 apical structure with other *Macromedaeus* species that the apical lobe wrapping the subdistal lobe; chelipeds of *M. orientalis* have blunt and concave finger tips like other *Macromedaeus*

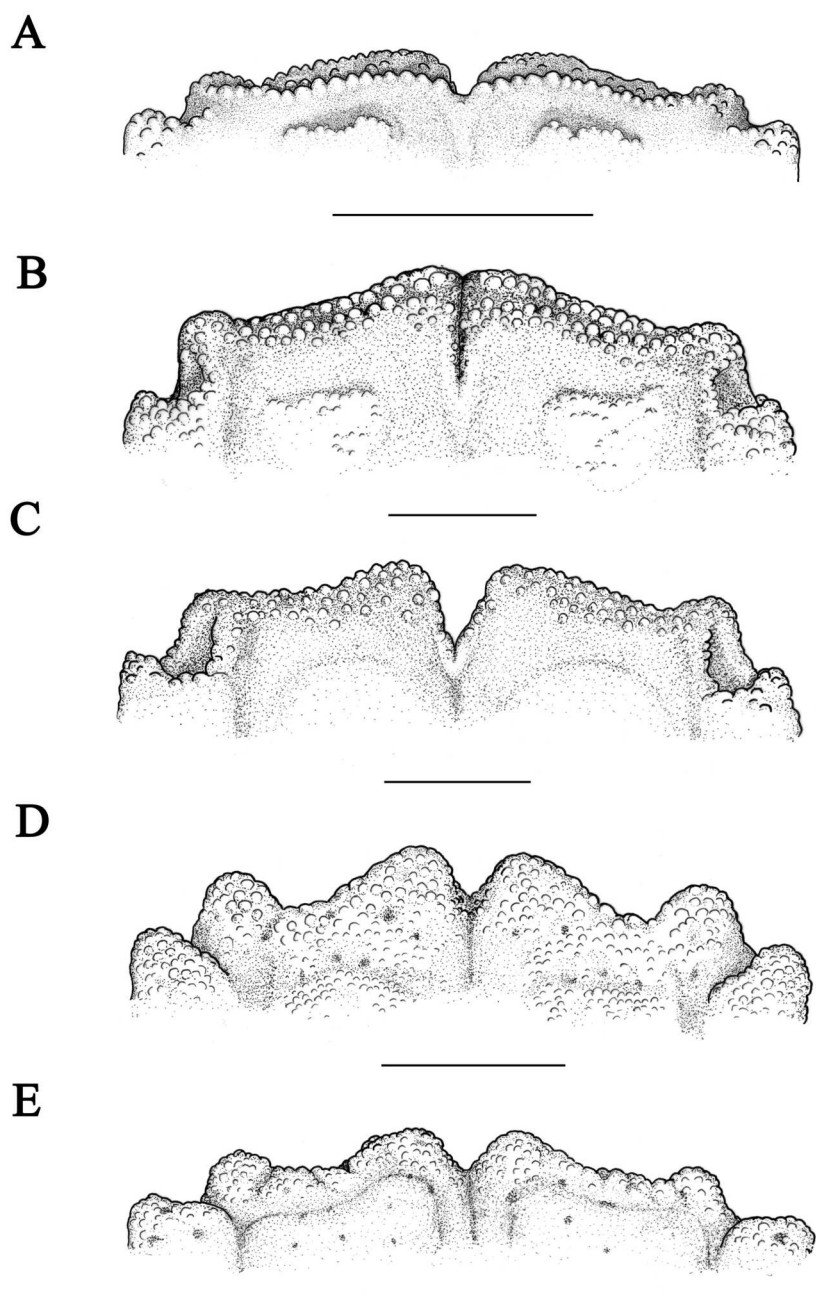

**Figure 5** **Front of *Macromedaeus* species, dorsal view.** (A) *M. orientalis* (*Takeda & Miyake, 1969*), male, 6.85 × 4.43 mm (MBM286994). (B) *M. distinguendus* (*De Haan, 1833-1850*), male, 27.76 × 18.21 mm (135CC03764). (C) *M. distinguendus* (*De Haan, 1833–1850*), male, 28.9 × 18.58 mm (MBM286999). (D) *M. quinquedentatus* (*Krauss, 1843*), male, 24.77 × 15.60 mm (MBM286997). (E) *M. crassimanus* (*Milne-Edwards, 1867*), male, 29.69 ×18.46 mm (MBM282426). Scale bars: (A) = 1 mm. (B-E) = 2 mm.

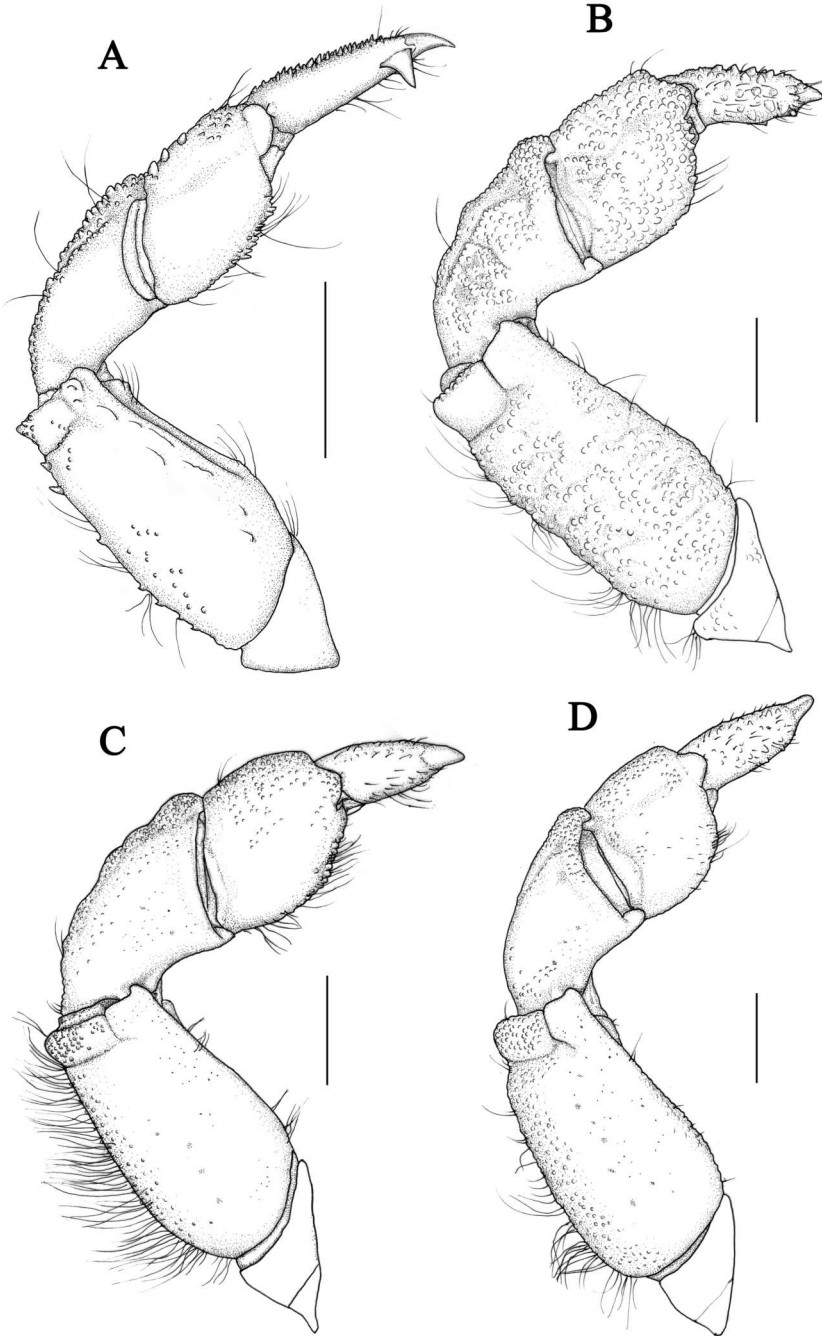

**Figure 6** **P5 of *Macromedaeus* species, dorsal view.** (A) *M. orientalis* (*Takeda & Miyake, 1969*), male, 6.85 × 4.43 mm (MBM286994). (B) *M. distinguendus* (*De Haan, 1833-1850*), male, 27.76 × 18.21 mm (135CC03764). (C) *M. quinquedentatus* (*Krauss, 1843*), male, 24.77 × 15.60 mm (MBM286997). (D) *M. crassimanus* (*Milne-Edwards, 1867*), male, 29.69 × 18.46 mm (MBM282426). Scale bars: (A) = 1 mm. (B–D) = 2 mm.

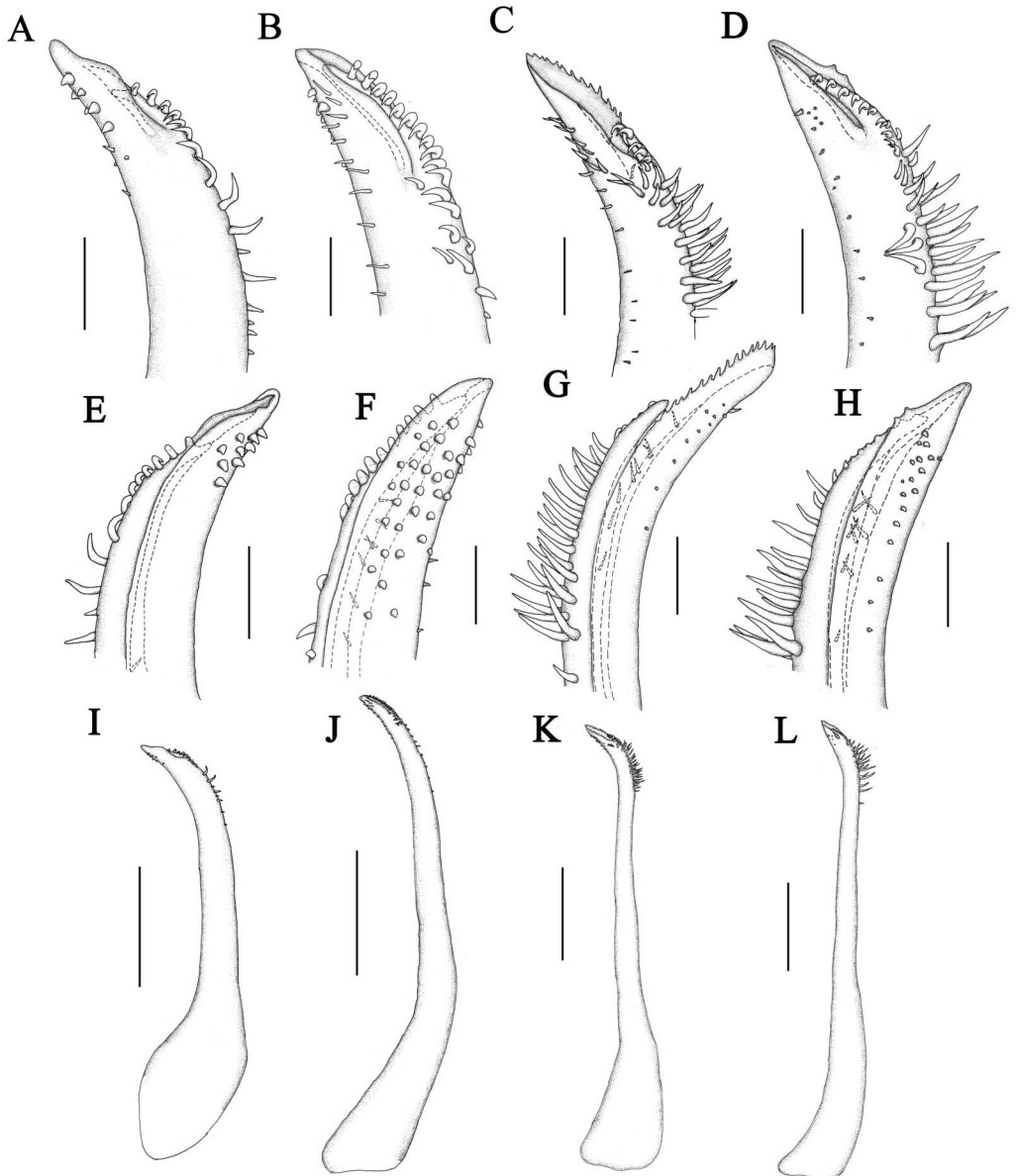

**Figure 7 Left G1 of *Macromedaeus* species.** (A, E, I) *M. orientalis* (*Takeda & Miyake, 1969*), male, 6.85 × 4.43 mm (MBM286994). (B, F, J) *M. distinguendus* (*De Haan, 1833-1850*), male, 27.76 × 18.21 mm (135CC03764). (C, G, K) *M. quinquedentatus* (*Krauss, 1843*), male, 24.77 × 15.60 mm (MBM286997). (D, H, L) *M. crassimanus* (*Milne-Edwards, 1867*), male, 29.69 × 18.46 mm (MBM282426). (A–D) distal portion, internal view. (E–H) same, external view. (I–L) overall, internal view. Scale bars: (A, B, E, F) = 0.1 mm. (C, D, E, H) = 0.2 mm. (I) = 0.5 mm. (J–K) = 1 mm.

species, while species of *Microcassiope* have more sharp and curving finger tips (*Guinot, 1967*).

    *M. orientalis* was firstly reported from China seas herein. The present specimens show some variations compared with previous descriptions in the following two characters: the front less prominent; G1 with an apical lobe slightly upturned on the tip and wrapping the subdistal lobe, ventral rim slightly lower than the dorsal rim (*Takeda & Miyake, 1969*; *Yamaguchi, Takeda & Tokudome, 1976*; *Lee & Ko, 2008*; *Lee, 2012*; *Maenosono, 2021*).

    *M. orientalis* is similar to *M. distinguendus. M. orientalis* differs from *M. distinguendus* by: smaller size; carapace rougher; front double-rimmed and divided into two lobes by a wide v-shape mesial notch deeply; anterolateral teeth narrower, apices more projecting; P5 slender, especially the propodus and dactylus, dactylus armed with a larger subterminal tooth, tip prolonged and curved backward and G1 apical lobe dorsal rim more prominent than the ventral rim (*Takeda, 1977*; *Maenosono, 2021*).

### *Macromedaeus distinguendus* (*De Haan, 1833-1850*)

*Cancer (Xantho) distinguendus De Haan, 1833-1850*: 48, pl. 13, fig. 7.
*Xantho distinguendus Alcock, 1898*: 113 (part); *Sakai, 1939*: 461, pl. 58, Fig. 4, pl. 191, fig.4; *Shen, 1932*, 97, fig. 56, 58a, b, pl. 2, fig. 5; *Gordon, 1931*: 543, figs. 21, 22c; *Forest & Guinot, 1961*: 57, fig.46; *Buitendijk, 1960*: 330; *Sendler, 1923*: 37.

    *Xanthodius distinguendus Balss, 1922*: 6
    *Chlorodius distinguendus Stimpson, 1858*: 34
    *Medaeus distinguendus De Man, 1887*: 31
    *Leptodius distinguendus Rathbun, 1931*: 100

    *Macromedaeus distinguendus Guinot, 1968*: 708; *Sakai, 1976*: 419, fig. 221, pl. 153, fig. 2; Dai et al., 1968: 265, fig. 151-1; 1991: 286, fig151-1, pl. 36, fig. 5; *Kim, 1973*: 379, fig. 143, pl. 27, fig. 108; *Serène, 1984*: 177 (keys); *Lee, 2012*: 154, figs. 118–120; *Ho, Yu & Ng, 2000*: 114 (list); *Ng et al., 2001*: 25 (list); *Ng et al., 2017*: 93 (list); *Liu, 2008*: 797 (list).
    Non *Xantho distinguendus Heller, 1861*: 323; *Alcock, 1898*: 113 (part); *Nobili, 1906a*; *Nobili, 1906b*: 239; *Laurie, 1906*: 401; *Klunzinger, 1913*: 200 (part); *Stebbing, 1918*: 51.

    = *Medaeops neglectus Balss, 1922*
    Non Xantho distinguendus *Klunzinger, 1913*: 203 (part), pl. 5, fig. 7
    = *Danielea noelensis Ward, 1935*
    Non Medaeus distinguendus *Henderson, 1893*: 359
    = *Medaeops neglectus Balss, 1922*

## Material examined

One male, one female; May. 8, 2010; Sanniang Bay, Qinzhou, Guangxi; Haiyan Wang et al. coll. (135CC03764); two males; Oct. 1, 2019; Lingshan Island, Qingdao, Shandong; Weiwei Xian coll. (MBM286999,); one male; May. 6, 2010; Hepu, Guangxi; Haiyan Wang et al. coll. (135CC03752); one male, May. 1, 2010; Shankou, Beihai, Guangxi, Haiyan Wang et al. coll. (135CC03760); six males, six females, Apr. 19, 2019; Rushan, Weihai, Shandong (MBM2867000); five males, one female; Jan. 27, 1951; Huiquan Bay, Qingdao, Shandong (MBM160961); two males; Sep. 16, 1955; Qingdao, Shandong (MBM160929); two males, one female; May. 21, 1963; Daheilan, Qingdao, Shandong; Xiubin Fang coll. (MBM160931); two males; May. 14, 1950; Qingdao, Shandong; (MBM160928); one male; Jul. 14, 1955; Zhonggang, Qingdao, Shandong; Zhengang Fan coll. (MBM160936); eight males, seven females, Dec. 25, 1950; east of Taipingjiao, Qingdao, Shandong; Xiuji Zhang coll. (MBM160926); two males; Aug. 14, 1957; Qingdao, Shandong; Yiqian Liu coll. (MBM160966); two males, four females; Sep. 22, 1955; Cangkou, Qingdao, Shandong, (MBM160940); 10 males, six females; Apr. 29, 1950; Qingdao, Shandong; Xiuji Zhang coll. (MBM160953); one male, one female ovigerous; Aug. 21, 1959; Zhonggang, Qingdao, Shandong; Zhongyan Qi, YongliangWang et al. coll. (MBM160941); seven males, one female ovigerous; Sep. 19, 1963; Daheilan, Qingdao, Shandong; Zhengang Fan et al. coll. (MBM160960); one male; Feb, 7, 1964; Huangdao, Qingdao, Shandong; Zhengang Fan et al. coll. (MBM160921); one male; Mar. 16, 1981; Hongshian, Qingdao, Shandong, Xianqiu Ren coll. (MBM160935); two females; Sep. 5, 1962; Zhonggang, Qingdao, Shandong (MBM160956); 14 males, six females; Oct. 4, 1979; Pingyang, Zhejiang; Xianqiu Ren coll. (MBM160943); two males, two females; Daheilan, Qingdao, Shandong, Zhengang Fan, Jieshan Xu coll. (MBM160942); one male; Nov. 17, 1963; Daheilan, Qingdao, Shandong; Zhengang Fan, Jieshan Xu coll. (MBM160946,); eight males, five females; Oct. 21, 1963; Daheilan, Qingdao, Shandong; Zhengang Fan et al. coll. (MBM160945); one male, one female; Nov. 2, 1956; Qingdao, Shandong; Zhengang Fan, Jieshan Xu coll. (MBM160939); two males; May. 22, 1951; Daheilan, Qingdao, Shandong; Xiuji Zhang coll. (MBM160923); three males; two females; Jun. 21, 1963; Daheilan, Qingdao, Shandong; Zhengang Fan coll. (MBM160938); one male, one female; Dec. 18, 1952; Qingdao, Shandong; Xiuji Zhang coll. (MBM160955); two males, two females; Aug. 24, 1980; Hongshian, Qingdao, Shandong (MBM160933); one male; Jul. 31, 1957; Taipingjiao, Qingdao, Shandong (MBM160927); one male; one female; May. 22, 1963; Changkou, Qingdao, Shandong; Xianqiu Ren, Xiubin Fang coll. (MBM160934); two males, three females; Sep. 9, 1983; Shijiusuo, Rizhao, Shandong (MBM160951); one male, one female; May. 25, 1975; Dongshan, Fujian; Mu Chen coll. (MBM160920); three males, three females; May. 29, 1957; Maidao, Qingdao, Shandong; Yiqian Liu coll. (MBM160959); two males, Oct. 4, 1979; Pingyang, Zhejiang (MBM160958); one female; Sep. 16, 1956; Qingdao, Shandong; (MBM160947); one male, two females; May. 31, 1957; Daheilan, Qingdao, Shandong; Jieshan Xu coll. (MBM160962); one female; Sep. 7, 1956; Maidao, Qingdao, Shandong; Zhengang Fan, Xiuji Zhang coll. (MBM160952); one female; Sep. 18, 1951; Qingdao, Shandong (MBM160957); eight males, one female; Sep. 16, 1955; Qingdao,

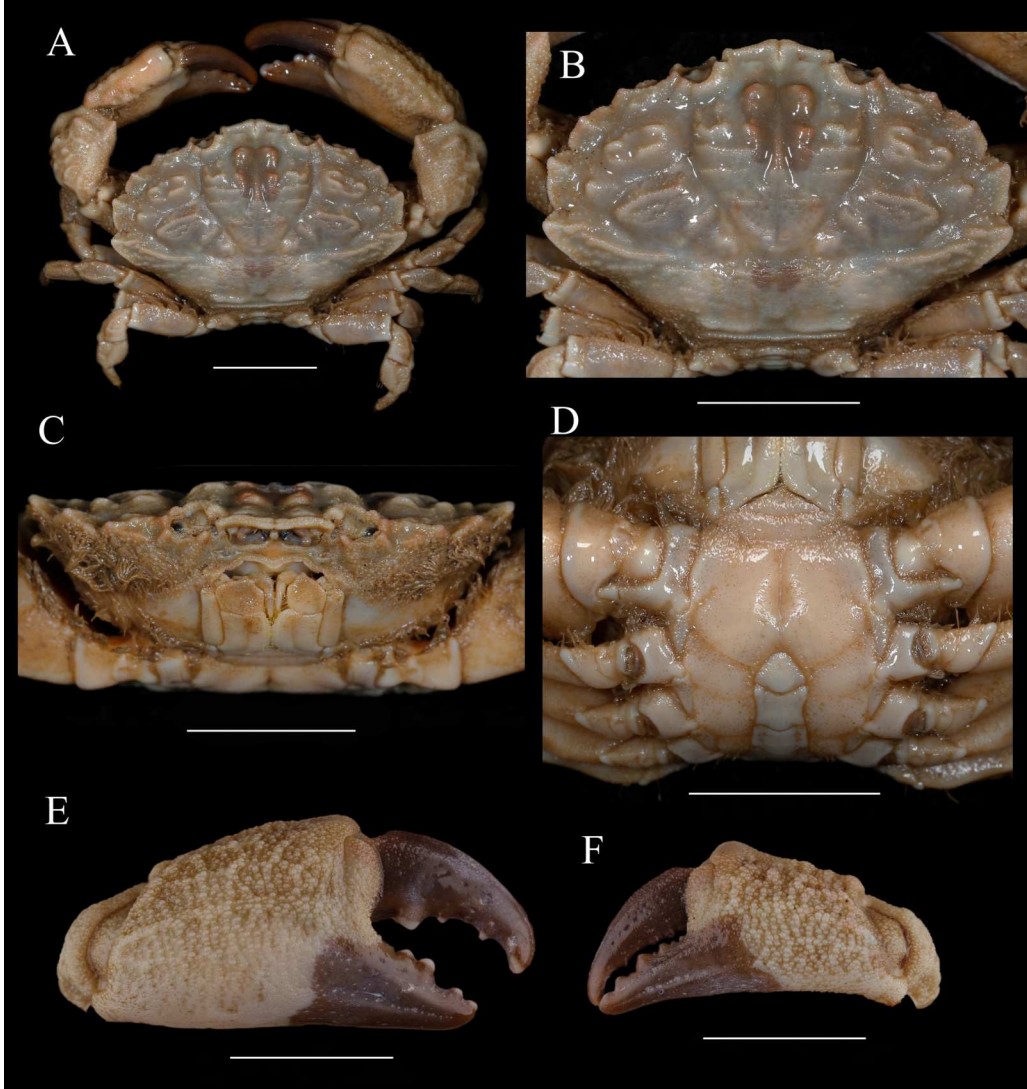

**Figure 8** *Macromedaeus distinguendus* (*De Haan, 1833-1850*), **male, 27.76 × 18.21 mm (135CC03764).** (A) Overall, dorsal view. (B) Carapace, dorsal view. (C) Third maxillipeds and pterygostomian region, anterior view. (D) Thoracic sternites and abdomen. (E) Right cheliped, outer view. (F) Left cheliped, outer view. Scale bars = 10 mm.

Shandong (MBM160925); five males, three females; Dec. 14, 1955; Daheilan, Qingdao, Shandong (MBM160954).

**Size:** CW: 6.74–30.58 mm, CL: 4.91–19.58 mm.

**Diagnosis.** Carapace (Figs. 8A, Fig. 8B) hexagonal, the breadth is about 1.5 times the length; dorsal surface granular, arranged in ridgy lines and crest; anterior 2/3 well defined, regions rise; front (Figs. 8B, 5B, 5C) narrow, about 1/4 the breadth of carapace, produced, edge grainy and oblique, middle produced, divided by a narrow crack. Dorsal orbital edge with double cracks, eyestalk with granules near the dorsal proximal border of cornea. anterolateral border armed with five teeth except the outer orbital angle; the first tooth

small; second and third broad, with an acute tip; the last tooth triangular. posterolateral border shorter than anterolateral border, slightly concave. Lateral surface of carapace with setae. Antennule situated transversely; orbital hiatus filled by antennal flagellum. Third maxilliped (Fig. 8C) completely covering buccal orifice; merus subquadrate, granulated; ischium subrectangular, with a smooth groove. Male thoracic sternites (Fig. 8D) smoother; groove between st1-2 and 3 slightly bending, median line of st4 deep.

Chelipeds (Figs. 8E, 8F) unequal; merus with setae on inner edge; carpus with a blunt tooth on inner angle, dorsal surface with irregular areoles and protuberances; palm covered with granules, irregular tubercles on dorsal outer surface; fingers black brown, the color of fixed finger extend to palm irregularly in male; dactylus with a central ridge and two longitudinal dorsal grooves, cutting edges with 4 blunt teeth; finger tips spoon-shaped.

Ambulatory legs (Figs. 8A, 6B) granular; merus anterior edges clothed with setae; carpus armed with three wavy protuberances on anterior edges, dactylus with spiny granules and setae, tip claw-shaped; posterior margin with spiny granules, without obvious subterminal tooth; dactylo-propodal articulation with lock composed of rounded prolongation on propodus distal lateral margin sliding against button on dactylus proximal lateral margin; P5 dactylus tip short, curved backward.

Abdominal somites (Fig. 8D) 3–5 completely fused in male; G1 (Figs. 7B, 7F, 7J) long and slender, slightly curving laterally; distal tip with an apical lobe wrapping the subdistal lobe; subdistal part with subterminal spines curve to dorsal, distal 11 of them on the prominent subdistal lobe; outer ventral surface of subdistal part with little spines.

Living color is shown in Fig. 9.

**Type locality.** Japan.

**Distribution.** Coast of China: Guangxi, Guangdong, Fujian, Taiwan, Zhejiang, Shandong Peninsula, Bohai Gulf; Mergui Archipelago, Korea, Japan, Tahiti, Palau.

**Remarks:** *M. distinguendus* is widely distributed along the coast of China. This species shows perceptible variations especially on: the granular crests on carapace, from low to rose; chelipeds dorsal surface of palm and carpus, from granular areolar to tubercular; anterolateral border teeth, from low and wide trilateral to prominent with sharp tip; the front divided by a narrow crack, or by a v-shape notch (Figs. 5B, Fig. 5C, 10). Those variations were identified as intraspecific.

The specimens identified by *Alcock (1898)*, *Heller (1861)*, *Nobili (1906a)*, *Nobili (1906b)*, *Klunzinger (1913)*, *Laurie (1906)* and *Stebbing (1918)* as *M. distinguendus* are actually *Medaeops neglectus* (*Balss, 1922*) (*Balss, 1922*; *Balss, 1924*; *Guinot, 1967*), meanwhile other specimens identified by *Klunzinger (1913)* as *M. distinguendus* are actually *Danielea*

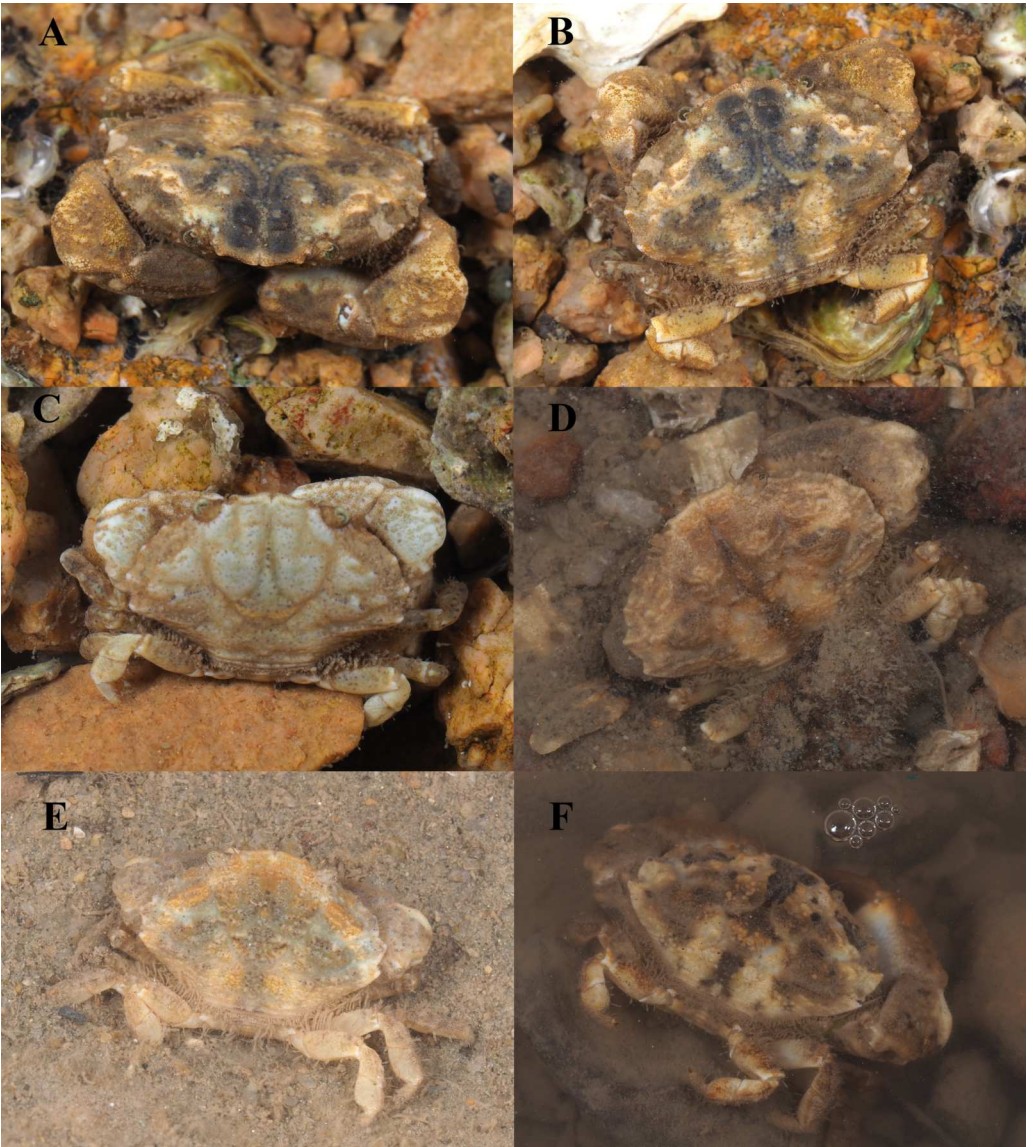

**Figure 9** *Macromedaeus distinguendus* (*De Haan, 1833-1850*). (A, B, C) From Qingdao, Shandong. (D) From Xiamen, Fujian. (E, F) From Ningbo, Zhejiang. Photo by Zhang Xu.

noelensis (*Ward, 1935*) (*Forest & Guinot, 1961*). Compared with *Medaeops neglectus*, *M. distinguendus* has chelipeds with spoon-shaped finger tips, and has distinct G1 morphology (*Balss, 1922*; *Balss, 1924*; *Odhner, 1925*; *Gordon, 1931*; *Forest & Guinot, 1961*).

*M. distinguendus* is similar with *M. hainanensis* sp. nov and *M. orientalis*. Diagnoses have been discussed in remarks of the latter two.

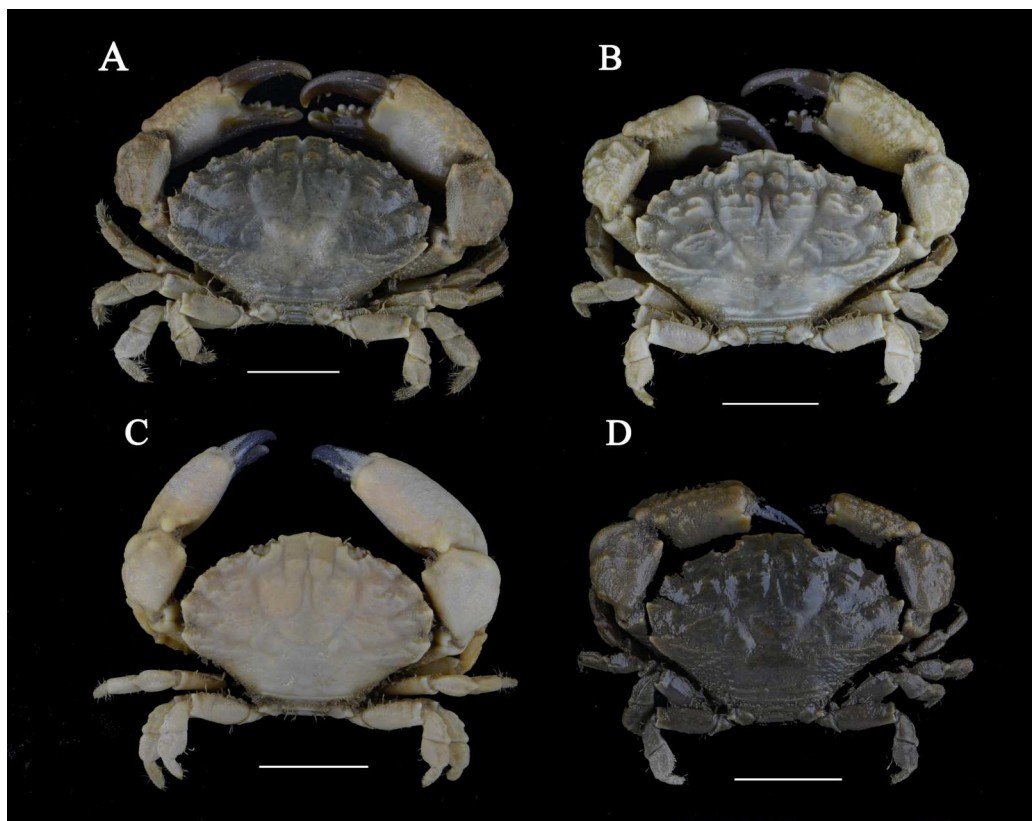

**Figure 10** *Macromedaeus distinguendus* (*De Haan, 1833-1850*). (A) Male, 28.9 × 18.58 mm (MBM286999). (B) Male, 27.76 × 18.21 mm (135CC03764). (C) Male, 23.19 × 15.75 mm (MBM160925). (D) Male, 24.14 × 16.04 mm (MBM160943). Scale bars = 10 mm.

### *Macromedaeus quinquedentatus* (*Krauss, 1843*)

*Cancer (Xantho) 5-dentatus Krauss, 1843*: 30, pl. 1 figs. 3a–3d
*Xantho (Leptodius) euglyptus Alcock, 1898*: 121
*Xantho (Leptodius) quinquedentatus Barnard, 1950*: 225, figs. 42f–42g
*Xantho quinquedentatus Buitendijk, 1960*: 321, figs. 9g–i
*Leptodius euglyptus Alcock & Anderson, 1899*: pl. 36 fig. 1
*Leptodius euglyptus quadrispinosus Chhapgar, 1957*: 429, pl. 9d–f

*Macromedaeus quinquedentatus Guinot, 1968*: 708; *Serène, 1984*: 177 (keys), 179, fig. 104, pl. XXV D-E; *Galil & Vannini, 1990*: 40; *Apel, 2001*: 87; *Mendoza, Lasley Jr & Ng, 2014*: 284; *Ghotbeddin & Naderloo, 2014*: 3, fig. 2b; *Naderloo, 2017*: 258, figs. 21, 32; *Maenosono, 2021*: 4, figs. 1b–c, 3a–g, 8b.

Non *Xantho quinquedentatus Edmondson, 1962*, 239, fig. 6d.

## Material examined

Two males; Mar. 23, 2008; Xiaodonghai, Hainan; Wei Jiang coll. (MBM282511); one male, one female; Dec. 24, 2007; Xiaodonghai, Hainan (MBM286995); one male; Dec. 25, 2007; Hainan; Wei Jiang coll. (MBM286996); two males; Jul. 5–6, 2020; Sanya, Hainan; Yunhao Pan, Fei Meng coll. (MBM286997); one male, one female; Nov. 11, 1990; Xiaodonghai, Hainan (MBM160944).

**Size:** CW: 5.84–24.77 mm, CL: 3.94–15.60 mm.

**Diagnosis:** Carapace (Figs. 11A, 11B) hexagonal, the breadth is about 1.5 times the length; dorsal surface finely granular; anterior 2/3 well-defined by broad and deep grooves. Front (Fig. 5D) narrow, about 1/5 the breadth of carapace, produced, divided into two lobes by a v-shape notch, edge grainy and sinuous, outer lobes wide and prominent, make a 4-lobes look. Dorsal orbital edge with double cracks, eyestalk with granules near the dorsal proximal border of cornea. anterolateral border armed with five teeth except the outer orbital angle; the first tooth blunt; second to fourth tooth about equal; the last tooth small. Posterolateral border concave, shorter than anterolatera. Setae present on lateral surface of carapace. Antennule situated transversely; antennal flagellum filling the orbital hiatus. Third maxilliped (Fig. 11C) completely covering the buccal orifice; merus subquadrate, granulated, with anterolateral angle produced; ischium subrectangular, with a groove. Male thoracic sternites (Fig. 11D) smooth, with sporadic granules; groove between st1-2 and 3 bending, st4 median line deep.

Chelipeds (Figs. 11E, 11F) unequal; merus inner edge with setae; carpus with an inner angle tooth, dorsal surface with irregular areoles and granules; palm prolonged, dorsal and outer surface covered with granules and tubercles, inner and ventral surface smooth; fingers black brown, in male individuals the color of fixed fingers extending to palm irregularly; dactylus curving, dorsal surface with double grooves, cutting edges with 4 triangular teeth; finger tips spoon-shaped.

Ambulatory legs (Figs. 11A, 6C) granular; merus anterior edges clothed with setae; carpus armed with three wavy protuberances on anterior edges; dactylus with spiny granules and setae, tip claw-shaped; sharp subterminal tooth present on dactylus posterior margin of the last two ambulatory legs in smaller individuals, only with granules in larger individuals; dactylo-propodal articulation with underdeveloped lock composed of rounded prolongation on propodus distal lateral margin sliding against lower button on dactylus proximal lateral margin; P5 dactylus tip curved backward.

Abdominal somites (Fig. 11D) 3–5 completely fused in male. G1 (Figs. 7C, 7G, 7K) long and slender, slightly curving laterally; distal tip with an elongate apical lobe wrapping the subdistal lobe; subdistal part with subterminal spines curve to dorsal, distal 6 of them on the prominent subdistal lobe; outer ventral surface of subdistal part with little spines; apical lobe with 5-16 sawtooth-like extensions on ventral brim.

**Type locality.** Natal bay, South Africa.

**Distribution.** Hainan Island; South Africa, Mozambique, Madagascar, Somalia, Golf of Aden, Dhofar, Gulf of Oman, Pakistan, India, Sri Lanka, Myanmar, Sulawesi Island, Flores Islands, Soela Islands, Talaud Islands, Timor Island, Misool Island, New Guinea, Okinawa.

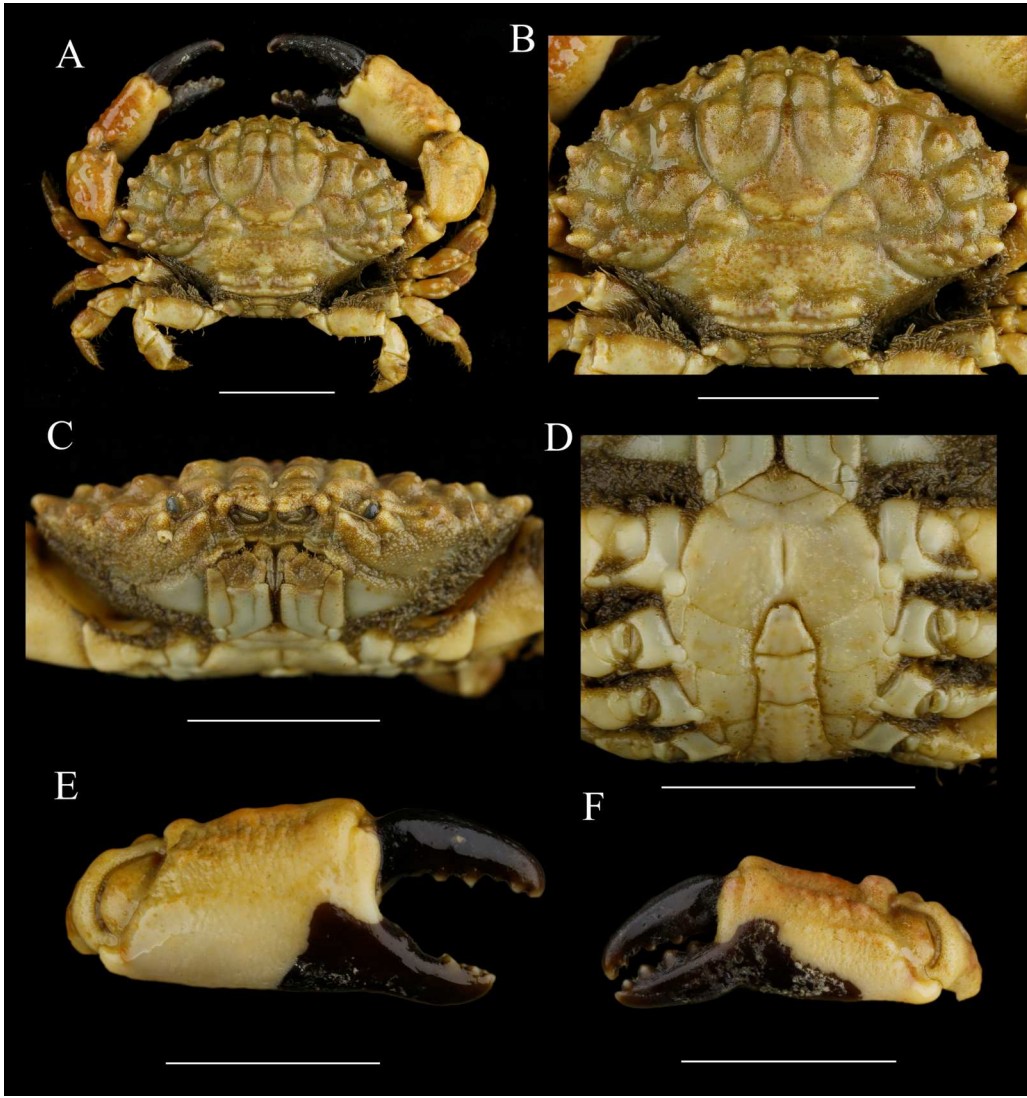

**Figure 11** *Macromedaeus quinquedentatus* (*Krauss, 1843*), **male, 24.77 × 15.60 mm (MBM286997).**
(A) Overall, dorsal view. (B) Carapace, dorsal view. (C) Third maxillipeds and pterygostomian region, anterior view. (D) Thoracic sternites and abdomen. (E) Right cheliped, outer view. (F) Left cheliped, outer view. Scale bars = 10 mm.

**Remarks.** *M. quinquedentatus* is newly recorded from China seas. G1 of the species with some degree of variation on the apical lobe length and sawtooth-like extensions number. In the present materials, the number of extensions is from five to 16 and the larger species generally have longer apical lobes and more extensions. Similar variations have been figured by *Buitendijk (1960)*.

The species is most related to *M. crassimanus* (*Milne-Edwards, 1867*) for having five anterolateral teeth and a sinuous front. *M. quinquedentatus* can be distinguished from *M. crassimanus* by: surface of carapace, chelipeds, ambulatory legs and third maxilliped with obvious granules, carapace slightly convex (surface with fine granules, carapace

obvious conves in *M. crassimanus*); fronto-orbital wide, only slightly less than half of the CW (fronto-orbital narrow, distinctly less than half the CW in *M. crassimanus*); carapace regions with areolae sharp, separated by large deep grooves (regions slightly projecting, separated by broad and shallow furrows in *M. crassimanus*); chelipeds palm armed with four prominent tubercles on superoexternal margin and with two longitudinal granulate ridges on external surface (only with rugose on superoexternal margin and a smoother external surface in *M. crassimanus*); G1 having a long apical lobe with more than five developed t extensions (G1 having a short apical lobe, with two very low extensions in *M. crassimanus*) (*Buitendijk, 1960*; *Serène, 1984*; *Mendoza, Lasley Jr & Ng, 2014*; *Maenosono, 2021*).

Specimens identified by *Edmondson (1962)* from Hawaii cannot be *M. quinquedentatus* now that it has a front without a distinct notch, carapace regions rise feebly and divided by narrow grooves and chelipeds without tubercles (*Maenosono, 2021*). Moreover, *Edmondson (1962)* indicated that the anterolateral border of the specimens was divided into 4 low teeth in addition to the external orbital angle, while *M. quinquedentatus* have 5 distinct anterolateral teeth behind the external orbital angle. Therefore, an examination of this specimen is necessary.

### *Macromedaeus crassimanus* (*Milne-Edwards, 1867*)
*Xantho crassimanus* *Milne-Edwards, 1867*: 267; *Rathbun, 1906*: 847; *Edmondson, 1925*: 51; *Edmondson, 1962*: 239, fig. 6c; *Buitendijk, 1960*: 318, figs. 9c–f.
    *Leptodius crassimanus* A. Milne-Edwards, 1873: 226, pl. 11, fig. 4; *De Man, 1888*: 287; *Forest & Guinot, 1961*: 63, fig. 48; *De Man, 1895*: 522;
    *Xantho exaratus var. crassimanus* *Ortmann, 1893*: 448
    *Xantho (Leptodius) crassimanus* *Alcock, 1898*: 118, 120; *Odhner, 1925*: 80.
    *Macromedaeus crassimanus* *Guinot, 1968*: 708, figs. 18, 22; *Takeda, 1976*: 85; *Serène, 1984*: 177 (keys), 179, fig. 103, pl. 25b; Dai et al., 1968: 266, figs. 151-2; *Dai & Yang, 1991*: 287, figs. 151-2, pl. 36, fig. 6; *Naderloo, 2017*: 257, fig. 21, 23; *Ho, Yu & Ng, 2000*: 114 (list); *Ng et al., 2001*: 25 (list); *Ng et al., 2017*: 93 (list); *Liu, 2008*: 797 (list); *Maenosono, 2021*: 5, figs 1d–e, 4a–g, fig. 8c.

### Material examined
One male; Nov. 24, 2007; Xiaodonghai, Hainan (MBM282426); one male; Jul. 5-7, 2020; Sanya, Hainan; Yunhao Pan, Fei Meng coll. (MBM286998).
    **Size.** CW: 20.29–29.69 mm, CL: 13.29–18.46 mm.
    **Diagnosis.** Carapace (Figs. 12A , 12B) transversely ovate, the breadth is about 1.6 times the length; dorsal rather convex, surface finely granular, with little pits; anterior 2/3 well-defined by broad and shallow grooves. Front (Fig. 5E) narrow, about 1/5 the breadth of Carapace, produced, divided into two lobes by a v-shape notch, edge grainy and sinuous, outer lobes wide and prominent make it a 4-lobes look. Dorsal orbital edge with double cracks, eyestalk with granules near the dorsal proximal border of cornea. Anterolateral border armed with five teeth except the outer orbital angle; the posterior two low; third and fourth wide and prominent, the last tooth small and acute. Posterolateral border

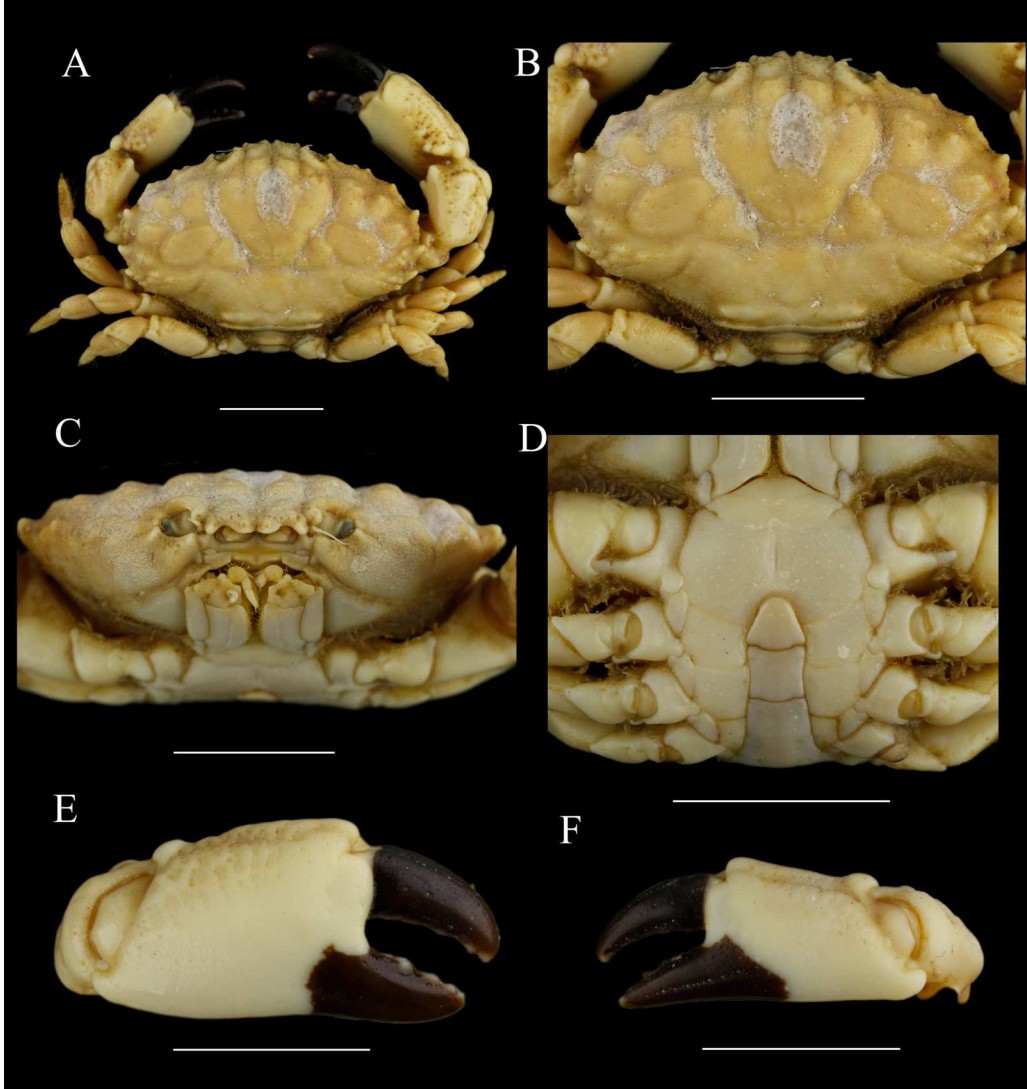

**Figure 12** *Macromedaeus crassimanus* (*Milne-Edwards, 1867*), **male, 29.69 × 18.46 mm (MBM282426).** (A) Overall, dorsal view. (B) Carapace, dorsal view. (C) Third maxillipeds and pterygostomian region, anterior view. (D) Thoracic sternites and abdomen. (E) Right cheliped, outer view. (F) Left cheliped, outer view. Scale bars = 10 mm.

concave, shorter than the anterolatera. Setae present on carapace lateral surface. Antennule transverse; antennal flagellum filling orbital hiatus. Third maxilliped (Fig. 12C) completely covering buccal orifice; merus subquadrate, with granules, anterolateral angle produced; ischium subrectangular, with a groove. Male thoracic (Fig. 12D) sternites smooth; groove between st1-2 and 3 bending, st4 median line deep.

Chelipeds (12E, 12F) unequal; merus with setae on inner edge; carpus with an inner angle tooth, dorsal surface granular; palm dorsal surface with irregular areoles and granules; fingers black brown, the fixed finger color extending to palm irregularly in male; dactylus

curving, with two longitudinal grooves dorsally, cutting edges armed with 4 triangular teeth; finger tips spoon-shaped.

Ambulatory legs (Figs. 12A, 6D) finely granular; anterior edges clothed with setae; carpus without obvious protuberances on anterior edges; dactylus with spiny granules and setae, tip claw-shaped, posterior margin with spiny granules, without subterminal tooth; dactylo-propodal articulation with underdeveloped lock composed of rounded prolongation on propodus distal lateral margin sliding against lower button on dactylus proximal lateral margin; P5 dactylus tip nearly straight, slightly forward pointed.

Abdominal somites (Fig. 12D) 3–5 completely fused in male. G1 (Figs. 7D, 7H, 7L) long and slender, slightly curving laterally; distal tip with an acute apical lobe wrapping the subdistal lobe; subdistal part with subterminal spines curve to dorsal, distal 4 of them on the prominent subdistal lobe; outer ventral surface of subdistal part with little spine; apical lobe with the ventral brim curling inward, and with two fine blunt extensions in the largest specimen.

**Type locality.** New Caledonia.

**Distribution.** Xisha islands, Hainan Island, Taiwan island; Red Sea, Persian Gulf, Pakistan, India, Sri Lanka, Andaman Islands, Sumatra, Java, Christmas Island, Sulawesi Island, Sabalana Islands, Flores Island, Morotai Island, Misool Island, Aru Islands, Tanimbar Islands, Maluku Islands, Timor Island, Palau, Australia, Japan, New Caledonia, Necker Island, Fiji, Hawaiian Islands, Line Islands, Samoa Islands, Tahiti Island.

**Remarks.** *Milne-Edwards (1867)* described *M. crassimanus* from New Caledonia and compared it with similar species. *M. crassimanus* is similar to *Leptodius sanguineus* (*Milne Edwards, 1834-1840*) and *Leptodius exaratus* (*Milne Edwards, 1834-1840*) morphologically, but can be identified by its sinuous, four-lobes like front, the chelipeds with fingers spoon-shaped but less hollow and the G1 with short apical lobe (*Milne Edwards, 1873*; *Buitendijk, 1960*). *M. crassimanus* is also related to *M. quinquedentatus*, the morphological differences have been discussed under remarks of the latter.

In the present largest specimen, there are two fine blunt extensions on the ventral brim of G1 apical lobe. Similar structure was figured by *Forest & Guinot (1961)*, fig. 48b) and *Dai et al. (1986)*, figs. 151-2, 1991, fig. 151-2), but replaced by smaller and sharper spines.

### *Macromedaeus demani* (*Odhner, 1925*)

*Xantho subacutus De Man, 1902*: 595, pl. 21 fig. 21
*Xantho demani Odhner, 1925*: 83 (part); *Balss, 1938*: 52; *Buitendijk, 1960*: 327, fig. 9i

*Macromedaeus demani Guinot, 1968*: 708; *Serène, 1984*: 177 (keys), pl. 25, fig. c; *Ho, Yu & Ng, 2000*: 114 (list); *Ng et al., 2001*: 25 (list); *Ng et al., 2017*: 93 (list); (*Liu, 2008*): 797 (list); (*Maenosono, 2021*): 7, figs. 1f-g, figs. 5a–f, figs. 6a–f, figs. 8d–e.

Non *Xantho demani Odhner, 1925*: 83 (part); *Ward, 1932*: 244
= *Lachnopodus subacutus Stimpson, 1858*

**Diagnosis.** *Buitendijk (1960)*, *Serène (1984)* and *Maenosono (2021)*.

**Distribution.** Taiwan Island; Lesser Soenda Islands, Ternate Island, Maluku Islands, Ryukyu Islands

**Remarks.** *Odhner (1925)* identified *X. demani* (synonym of *M. demani*) from specimens that identified as *Lachnopodus bidentatus* (*Milne-Edwards, 1867*) by *Alcock (1898)* from Andaman Islands and *Lachnopodus subacutus* (*Stimpson, 1858*) by *De Man (1902)* from Ternate. However, *Forest & Guinot (1961)* did not accept the identification of *Odhner (1925)*, and believed *M. demani* was not an available species. *Forest & Guinot (1961)* also examined part of specimens of *Ward (1932)* identified as *X. demani* collected from Capricorn Group, Queensland, and listed these materials in the synonymia of *L. subacutus*. *Buitendijk (1960)* examined the materials collected from Ternate and Todore, and approved the availability of *X. demani*. *Buitendijk (1960)* also listed the differences between this species and *L. subacutus,* attaching with the G1 figure of *X. demani* which had never been figured before. *Maenosono (2021)* examined specimens of *M. demani* in similar size with the materials of *De Man (1902)*, and supported that the specimens described by *De Man (1902)* should be *M. demani* rather than *L. subacutus.* The suggestion of *Forest & Guinot (1961)* have been widely accepted, so some authors listed *M. demani* materials of *De Man (1902)*, *Odhner (1925)* or *Buitendijk (1960)* in synonymia of *L. subacutus,* mainly *Forest & Guinot (1961)*, *Takeda (1976)* and Dai & Yang (*1986*; *1991*), what caused confusions (*Maenosono, 2021*).

The most striking differences between *M. demani* and *L. bidentatus* or *L. subacutus* are: chelipeds with finger tips spoon-shaped in *M. demani* (it sharp in *L. bidentatus* and *L. subacutus*); G1 with apical lobe wraping the subdistal lobe in *M. demani* (it with two sharp apical lobes in *L. bidentatus* and *L. subacutus*) (*Buitendijk, 1960*; *Maenosono, 2021*).

### Molecular data analyses

Molecular phylogenetic analyses were performed on 16 species within 7 genera (Table 1). BI and ML trees of the combined data produce similar structures. BI posterior probabilities (PP) and ML bootstrap replications (BS) were labeled (Fig. 13).

The present phylogenetic analyses indicated that the genus *Microcassiope* was located at the base part of the Xanthidae and not associated with the species of *Macromedaeus* (PP100/BS100), making Xanthinae polyphyletic. Two genera, *Macromedaeus* and *Leptodius* were identified as sister taxa (PP100/BS100), in accordance with the morphological conclusion. *Etisus* was not a monophyly taxon now that *Etisus sakaii* (*Takeda & Miyake, 1968*) associated with *Macromedaeus* and *Leptodius* (PP80/BS-). G1 of the species was characterized with mushroom-like extensions on the ventral brim of apical lobe and curving spine on the subdistal lobe (see *Takeda & Miyake,*
**Table 1** Species and sequences used in the phylogenetic analysis with GenBank accession numbers and source.

| Species | Location | Voucher ID | GenBank accession number | | | | | Sources |
|---|---|---|---|---|---|---|---|---|
| | | | **12S** | **16S** | **18S** | **COI** | **H3** | |
| *Macromedaeus crassimanus* (*Milne-Edwards, 1867*) | Xiaodonghai, Sanya, Hainan, China | MBM282426 | MZ901149 | MZ901133 | MZ901166 | N/A | N/A | Present study |
| *M. crassimanus* (*Milne-Edwards, 1867*) | Balicasag Island, Philippines | ZRC 2003.0369 | N/A | N/A | N/A | HM751018 | N/A | *Lai et al. (2011)* |
| *M. crassimanus* (*Milne-Edwards, 1867*) | India | CASMBGM-1MC | N/A | N/A | N/A | MG725244 | N/A | Unpublished |
| *M. distinguendus* (*De Haan, 1833-1850*) | Hepu, Guangxi, China | 135CC03752 | MZ901151 | MZ901135 | MZ901167 | N/A | MZ908676 | Present study |
| *M. distinguendus* (*De Haan, 1833-1850*) | Sanniangwan, Guangxi, China | 135CC03764 | MZ901152 | MZ901136 | MZ901168 | MZ900931 | MZ908677 | Present study |
| *M. distinguendus* (*De Haan, 1833-1850*) | Weihai, Shandong, China | MBM287000 | MZ901159 | MZ901141 | MZ901173 | MZ900938 | MZ908684 | Present study |
| *M. distinguendus* (*De Haan, 1833-1850*) | South Korea | DeB39 | N/A | N/A | N/A | JX502906 | N/A | Unpublished |
| *M. distinguendus* (*De Haan, 1833-1850*) | South Korea | DeB41 | N/A | N/A | N/A | JX502907 | N/A | Unpublished |
| *M. distinguendus* (*De Haan, 1833-1850*) | South Korea | DeB43 | ”-”-N/A | N/A | N/A | JX502908 | N/A | Unpublished |
| *M. hainanensis* sp. nov. | Mulan Bay, Wenchang, Hainan, China | MBM286989 | MZ901145 | MZ901129 | MZ901162 | MZ900926 | MZ908671 | Present study |
| *M. hainanensis* sp. nov. | Mulan Bay, Wenchang, Hainan, China | MBM286990 | MZ901147 | MZ901131 | MZ901164 | MZ900928 | MZ908673 | Present study |
| *M. orientalis* (*Takeda & Miyake, 1969*) | Hainan, China | MBM286993 | MZ901146 | MZ901130 | MZ901163 | MZ900927 | MZ908672 | Present study |
| *M. quinquedentatus* (*Krauss, 1843*) | Xiaodonghai, Sanya, Hainan, China | MBM286995 | MZ901150 | MZ901134 | N/A | MZ900930 | MZ908675 | Present study |
| *Chlorodiella laevissima* (*Dana, 1852*) | Hainan, China | MBM287001 | MZ901148 | MZ901132 | MZ901165 | MZ900929 | MZ908674 | Present study |
| *C. nigra* (Forskål, 1775) | Hainan, China | MBM287002 | MZ901142 | MZ901126 | MZ901160 | MZ900923 | MZ908668 | Present study |
| *C. nigra* (Forskål, 1775) | Sanya, Hainan, China | MBM287003 | MZ901143 | MZ901127 | MZ901161 | MZ900924 | MZ908669 | Present study |
| *Etisus anaglyptus* H. Milne Edwards, 1834 | Weizhou Island, Beihai, Guangxi, China | 135CC06153 | MZ901155 | MZ901138 | MZ901170 | MZ900934 | MZ908680 | Present study |
| *E. laevimanus* Randall, 1840 | Lingchang reef, Lingao, Hainan, China | 135CC06297 | MZ901156 | MZ901139 | MZ901171 | MZ900935 | MZ908681 | Present study |
| *E. laevimanus* Randall, 1840 | Lingchang reef, Lingao, Hainan, China | 135CC06301 | MZ901157 | MZ901140 | MZ901172 | MZ900936 | MZ908682 | Present study |
| *E. sakaii* (*Takeda & Miyake, 1968*) | Sanya, Hainan, China | MBM287004 | MZ901158 | N/A | N/A | MZ900937 | MZ908683 | Present study |
| *Leptodius affinis* (*De Haan, 1833-1850*) | Dachan reef, Danzhou, Hainan, China | 135CC06140 | MZ901153 | N/A | MZ901169 | MZ900932 | MZ908678 | Present study |
| *L. affinis* (*De Haan, 1833-1850*) | Hainan, China | 135CC06145 | MZ901154 | MZ901137 | N/A | MZ900933 | MZ908679 | Present study |
| *L. davaoensis* Ward, 1941 | Fengjiawan, Wenchang, Hainan, China | MBM286757 | MZ901144 | MZ901128 | N/A | MZ900925 | MZ908670 | Present study |
| *Microcassiope taboguillensis* (*Rathbun, 1906*) | Cohiba Island, Panama | ULLZ 11881 | KF683035 | KF682967 | KF682854 | KF682825 | KF682573 | *Thoma, Danièle & Felder (2014)* |
| *Microcassiope xantusii* (Stimpson, 1871) | Cohiba Island, Panama | ULLZ 11880 | KF683036 | KF683006 | KF682853 | KF682827 | KF682546 | *Thoma, Danièle & Felder (2014)* |
| *Eriphia gonagra* (Fabricius, 1781) | Ft. Pierce, Florida, USA | ULLZ 5463 | HM637933 | HM637964 | HM637998 | HM638035 | HM596633 | *Lai et al. (2011)* |
| *Menippe rumphii* (Fabricius, 1798) | Labrador Beach, Singapore | ZRC 2003.0211 | HM637946 | HM637976 | HM638015 | HM638051 | HM596626 | *Lai et al. (2011)* |

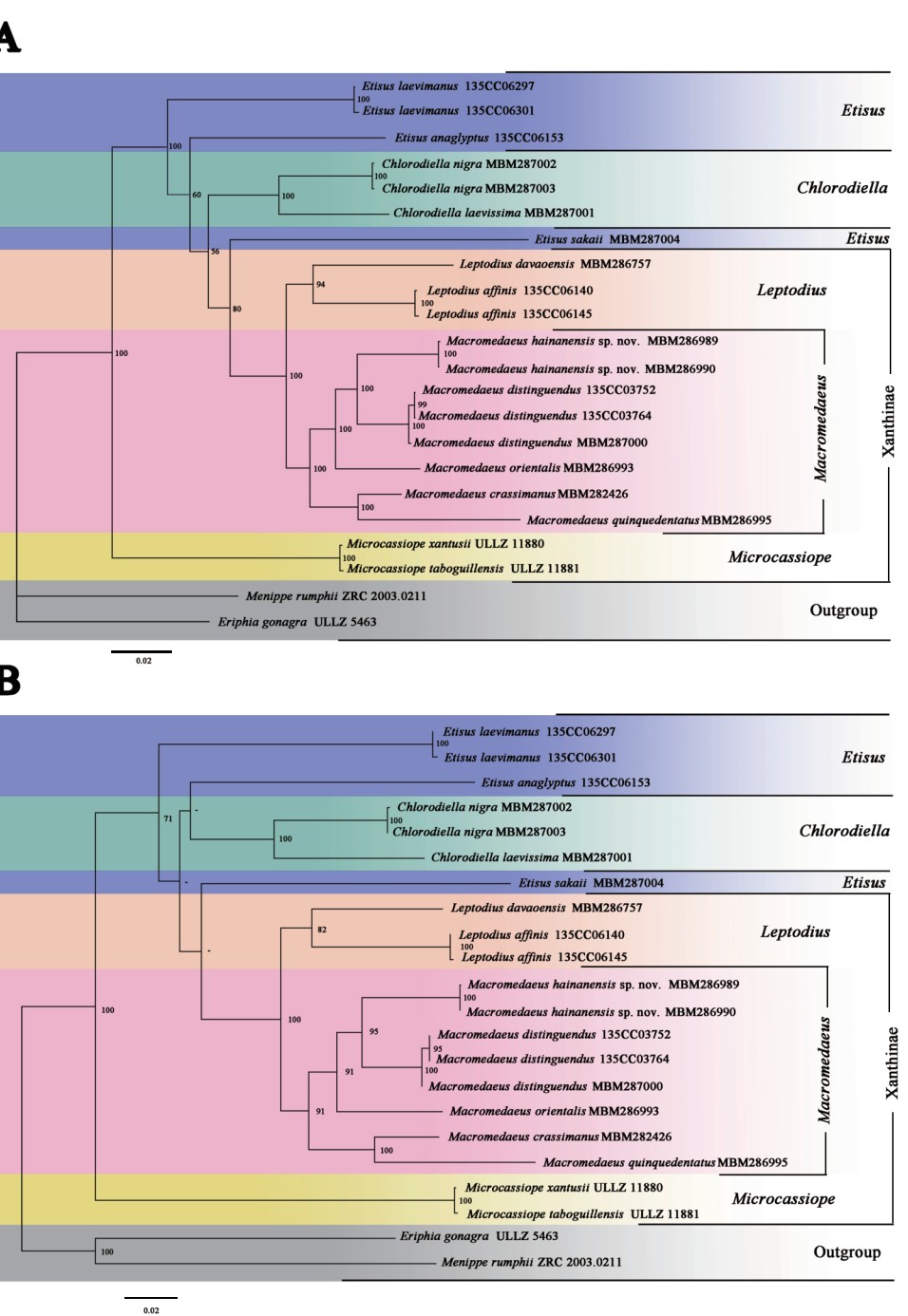

**Figure 13** **Phylogenetic relationships inferred from combined 12S, 16S, 18S, COI and H3 sequences among *Macromedaeus* species and related genus in Xanthidae, analyzed by Bayesian Inference (BI) and maximum likelihood (ML) analyses.** (A) BI tree, with posterior probabilities (PP) labeled. (B) ML tree, with bootstrap replications (BS) labeled, "-" represents values below 50.

**Table 2  Genetic divergence of COI gene among the five species of Macromedaeus calculated from Kimura 2-Parameter-coreected calculations.**

|  | *M. crassimanus* | *M. distinguendus* | *M. orientalis* | *M. quinquedentatus* | *M. hainanensis* sp. nov. |
|---|---|---|---|---|---|
| *M. crassimanus* | 0 | | | | |
| *M. distinguendus* | 0.133 | 0.000675106 | | | |
| *M. orientalis* | 0.131 | 0.098 | N/A | | |
| *M. quinquedentatus* | 0.124 | 0.133 | 0.135 | N/A | |
| *M. hainanensis* sp. nov. | 0.130 | 0.097 | 0.109 | 0.136 | 0.001687764 |

*1968*, figs. 3a–c). Similar morphological characteristics are present on *Leptodius* and Mac 2, both of which had the curving subdistal spines and mushroom-like, tongue-like or sawtooth-like extensions on the ventral brim of G1 apical lobe. In addition, the status of *E. anaglyptus* H. Milne Edwards, 1834 was slightly different in ML and BI trees.

Five species of genus *Macromedaeus* formed a monophyly clade with high support values (PP100/BS91). The genetic divergences of COI between *M. hainanensis* sp. nov. and other four congeneric species were higher than 0.09 (Table 2). These species can be further divided into two clades, Mac 1 (PP100/BS95) and Mac 2 (PP100/BS100), which were well supported. Mac 1 contained the new species *M. hainanensis* sp. nov., *M. orientalis* and *M. distinguendus*. They share similar morphological characteristics, including 4 anterolateral teeth, front with not produced and narrow outer lobes, G1 more curved, the subdistal part with larger short spines on the outer surface, while sparser long spines on the inner surface, and apical lobe of G1 without extensions on ventral brim. Mac 2 contained *M. quinquedentatus* and *M. crassimanus.* They shared the following morphological characteristics: 5 anterolateral teeth, front with produced and wider outer lobes, make a 4-lobes look, G1 more direct, the subdistal part with smaller short spines on the outer surface, while denser long spines on the inner surface, and apical lobe of G1 with obvious or fuzzy extensions on ventral brim.

## DISCUSSION

In the present study, a new species, *M. hainanensis* sp. nov., is described. *M. hainanensis* sp. nov. was identified from other *Macromedaeus* species by morphological characteristics and phylogenetic analyses. Two newly recorded species of Chinese waters, *M. orientalis* and *M. quinquedentatus* are reported. The controversial species *M. orientalis* was classified in *Macromedaeus* based on morphological and phylogenetic evidence. Geographical distributions of these species were summarized. A key of *Macromedaeus* species was provided. Consideration of the obvious variation on the carapace, the morphology of G1, front and P5 should be better diagnostic characteristics for *Macromedaeus*.

The phylogenetic analyses confirmed the monophyly of *Macromedaeus* based on species involved. *Macromedaeus* can be divided into two clades, Mac1 and Mac2. Species of the same clade share similar characteristics that are consistent with the morphological results. However, the *Etisus* and *Xanthinae* were not monophyletic

in the present study. *Microcassiope* located near the base of Xanthidae in our trees and similar topological structure shown in the phylogenetic research of *Thoma, Danièle & Felder (2014)*. *E. sakaii* is associated with the clade of *Macromedaeus* and *Leptodius*. It shares similar G1 form with part of species in this clade, that with mushroom-like or tongue-like extensions. Further study will make sure is this characteristic has broader phylogenetic significance, now that similar structure also known in other Xanthidae species such as *Etisus electra* (*Herbst, 1801*), *Etisus frontalis* (*Dana, 1852*) *Etisus demani* (*Odhner, 1925*), and *Liocarpilodes armiger* (*Nobili, 1906a*; *Nobili, 1906b*).

## Key to species of *Macromedaeus Ward, 1942* (adapted from *Serène (1984)* and *Maenosono (2021)*

1. Anterolateral border with about ten short, irregular teeth with obtuse apices. The surface of the carapace and chelipeds is irregularly hollowed giving it a reticulated appearance; P5 dactylus with spiny granules, larger in terminal; G1 see *Serène, 1984*, fig.101.......................................................................................... *M. nudipes*

–Anterolateral border with 4–five teeth or lobes...........................................................................................2

2. Anterolateral border with five teeth, front sinuous, with produced and wide outer lobes, make a 4-teeth look.............................. ......................................................................................................................................3

–Anterolateral border with four teeth, front with not produced and narrower outer lobes..................................4

3. Carapace regions with areolae sharper, separated by large deep grooves; chelipeds palm armed with prominent tubercles on superoexternal margin; G1 with 5–16 sawtooth-like extensions on ventral brim.................................................................................................................................... *M. quinquedentatus*

–Carapace regions slightly projecting, separated by broad, shallow furrows; chelipeds only with rugose on superoexternal margin, external surface smoother; G1 with the ventral brim curling inward with very fine blunt extensions.................................................................................................. *M. crassimanus*

4. Anterolateral border with 4 feeble lobes, first two are sometimes not easily distinguished one from the other; surface of carapace and chelipeds have small hollows, giving them a slightly rugose aspect; P5 dactylus with granules, larger in terminal; G1 see *Buitendijk, 1960* fig. 9j........................................................ *M. demani*

–Anterolateral border with 4 distinct teeth; surface of carapace and chelipeds with granules.........................5

5. Carapace regions noticeably projecting, separated by deeper and more acute furrows; anterolateral teeth with noticeably projecting apices forming an acute angle; surface of carapace and chelipeds is irregularly granular. the granules are much larger and more acute on the frontal margin, tantero-lateral teeth and neighboring regions, superoexternal parts of the carpus and propodus of the chelipeds and the anterior margins of the ambulatory legs; rows of transverse, irregular, acute granules occur here and there on the carapace regions. G1 see *Serène, 1984*, fig. 102................................................................. *M. voeltzkowi*

–Carapace regions hardly projecting, separated by feeble furrows.....................................................................6

6. Front produced, not double-rimmed; anterolateral teeth broader, apices slightly projecting, forming an obtuse angle; ambulatory legs carpus with three crests, dactylus without distinct subterminal tooth.............................................................................................................................. *M. distinguendus*

–Front not so produced, double-rimmed; anterolateral teeth narrower, apices projecting forming an acute angle; ambulatory legs carpus without obvious crests, a sharp and strong subterminal tooth on dactylus of last three legs.......................................................................................................................................................7

7. P5 slender, especially the obvious prolonged propodus and dactylus; dactylus tip prolonged, curved backward; the color of fixed finger extending to palm irregularly in male chelipeds; G1 more stout, distal tip with a apical lobe slightly upturned, the apical lobe dorsal rim more prominent than the ventral rim.................................................................................................................................*M. orientalis*

–P5 stout, dactylus tip stout, nearly straight, slightly pointed forward; the colour of fixed finger nearly not extend to palm in chelipeds; G1 more slender, apical lobe not upturned, the ventral rim more prominent than the dorsal rim.......................................................................................................... *M. hainanensis* sp. nov.

## CONCLUSIONS

A new species, *M. hainanensis* sp. nov. was described from the South China Sea. *M. hainanensis* sp. nov. is most similar to *M. distinguendus* and *M. orientalis*, but can be distinguished by morphological and molecular evidence. *M. orientalis* and *M. quinquedentatus* were reported from Chinese waters for the first time, and the six *Macromedaeus s* pecies present recorded from Chinese waters were reviewed using integrative taxonomy methods. *M. orientalis* should be transferred to *Macromedaeus* instead

of *Microcassiope*. The present analysis supports the monophyletic of *Macromedaeus*, which can be further divided into two clades well supported, but *Etisus* and Xanthinae are not monophyletic. Further phylogenetic studies involving more taxa are helpful to clarify the status of other Xanthidae crabs.

# ACKNOWLEDGEMENTS

We would like to thank Yang Li, Fei Meng, Yunhao Pan, Xianqiu Ren, Haiyan Wang, Weiwei Xian, Julong Zhang, Shuqian Zhang and other scientists and participants for their help in collecting materials. We are also thankful to Xu Zhang for providing photographs of living crabs used in this paper.

## Funding

This work was supported by the National Natural Science Foundation of China (42176138), the National Key R&D Program of China (2018YFD0900804), the National Science & Technology Fundamental Resources Investigation Program of China (2018FY10010006) and the National Science Foundation for Distinguished Young Scholars (42025603). The funders had no role in study design, data collection and analysis, decision to publish, or preparation of the manuscript.

## Grant Disclosures

The following grant information was disclosed by the authors:
National Natural Science Foundation of China: 42176138.
National Key R&D Program of China: 2018YFD0900804.
National Science & Technology Fundamental Resources Investigation Program of China: 2018FY10010006.
National Science Foundation for Distinguished Young Scholars: 42025603.

## Competing Interests

The authors declare there are no competing interests.

## Author Contributions

- Ziming Yuan conceived and designed the experiments, performed the experiments, analyzed the data, prepared figures and/or tables, authored or reviewed drafts of the paper, and approved the final draft.
- Wei Jiang and Zhongli Sha conceived and designed the experiments, authored or reviewed drafts of the paper, and approved the final draft.

## Data Availability

The DNA sequences are available in the Supplementary Files and at GenBank: COI sequences: MZ900923 to MZ900938, 16S sequences: MZ901126 to MZ901141, 12S sequences: MZ901142 to MZ901159, 18S sequences: MZ901160 to MZ901173 and H3 sequences: MZ908668 to MZ908684.

## New Species Registration

The following information was supplied regarding the registration of a newly described species:

Publication LSID: urn:lsid:zoobank.org:pub:07834F78-CD9D-4516-A92C-7FC5B680A7F9.

*Macromedaeus hainanensis* LSID: urn:lsid:zoobank.org:act:C7583031-355B-41F9-B219-B98652F06A17.

## Supplemental Information

Supplemental information for this article can be found online at http://dx.doi.org/10.7717/peerj.12735#supplemental-information.

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
