# Peer review of "A review of the common crab genus *Macromedaeus* Ward, 1942 (Brachyura, Xanthidae) from China Seas with description of a new species using integrative taxonomy methods"

_PeerJ, doi:10.7717/peerj.12735_

## Round 0.1 · original submission · Major Revisions

As commented by the reviewers, I agree that the manuscript, although it contains sufficient data that would contribute to the field of brachyuran taxonomy and species diversity, would benefit greatly from a more detailed description and elaboration, especially the abstract section. A careful review of the revised manuscript regarding the consistency and grammar of the overall language of the manuscript. On a side note, the high-quality figures provided by the authors are commendable.

Reviewer 1 ·

Basic reporting

Before all, the text requires much elaboration, which should be made before formal submission.

Experimental design

See above.

Validity of the findings

The recognition of the new species seems to be warranted by genetic and morphological evidence.

Annotated reviews are not available for download in order to protect the identity of reviewers who chose to remain anonymous.

Reviewer 2 ·

Basic reporting

TThe MS deals with a taxonomically very complicated genus of brachyuran crabs. Its well-structured and clear written. It describes a new species, which I think is a good distinct specie. The drawings and photos are in high quality condition.

But there are some comments, which I propose to be applied before get it published

1. Abstract is very short and vague. Please provide some details of your result and conclusion. Your abstract has actually no data, just some vague citation. You can give name of your new records, amount of recorded species of the genus some phylogenetic and biogeographical information.

2. The English of the MS must be checked and improved. For instance, in the first line of Introduction "The genus Macromedaeus Ward, 1942, currently including six described species found from Indo-West Pacific"

"including" must be "includes"
"Indo-West Pacific" must be "the Indo-West Pacific"

3. There is no mention about the state of propodus dactylus articulation in M. orientalis.

4. The subterminal spine of ambulatory legs, usually, is present in the species with propodus-dactylus articulation. What is the state of this key (subterminal spine) character in other species? and what the implications might be?

5. The syntypes of M. demani are juvenile and the detailed morphology of G1 are not distinguished. In addition, one of the characteristics of M. demani is lack of anterolateral teeth and presence of anterolateral lobes–based on serene (1984). How can we be sure that the characters stated in the manuscript belong to M. demani and not from a new species? Given the characteristics of the syntype M. demani slightly differ from those of the specimen in the manuscript.

6. There is no taxonomical account on M. crassimanus.

7. The taxonomical account of M. demani is the iteration of what Serene (1984) and Guinot (1968) stated. Additionally, the authors mention that some authors like serene (1984) list M. demani as a synonym of L. subacutus. However, Serene (1984) listed M. demani a separated species.

Experimental design

no comment

Validity of the findings

no comment

Additional comments

no comment

---

## Round 0.2 · Minor Revisions

The revised manuscript is very much improved. However, I still have some concerns that I think will aid in improving the manuscript as follows:

Line 17-18: "M. hainanensis is most related to M. distinguendus (De Haan, 1835) and M. orientalis", based on?
Line 112: Please also describe where and how the samples were coltected. and When?
Line 126: Please include, in bracket, the kit producer country of origin.
Line 135: PCR 'was'
line 160: each dataset 'was'

If possible, I would like to request the authors to kindly proofread the whole manuscript again for any minor grammatical errors.

---

## Round 0.3 · accepted · Accept

The language of the manuscript has significantly improved. I believed that the results of the study by Yuan et al. would greatly contribute to the brachyuran taxonomy, especially in the Asia region.